# Distinct interactions of eIF4A and eIF4E with RNA helicase Ded1 stimulate translation in vivo

**Suna Gulay, Neha Gupta, Jon R Lorsch, Alan G Hinnebusch***

Division of Molecular and Cellular Biology, Eunice Kennedy Shriver National Institute of Child Health and Human Development, National Institutes of Health, Bethesda, United States

**Abstract** Yeast DEAD-box helicase Ded1 stimulates translation initiation, particularly of mRNAs with structured 5'UTRs. Interactions of the Ded1 N-terminal domain (NTD) with eIF4A, and Ded1-CTD with eIF4G, subunits of eIF4F, enhance Ded1 unwinding activity and stimulation of preinitiation complex (PIC) assembly in vitro. However, the importance of these interactions, and of Ded1-eIF4E association, in vivo were poorly understood. We identified separate amino acid clusters in the Ded1-NTD required for binding to eIF4A or eIF4E in vitro. Disrupting each cluster selectively impairs native Ded1 association with eIF4A or eIF4E, and reduces cell growth, polysome assembly, and translation of reporter mRNAs with structured 5'UTRs. It also impairs Ded1 stimulation of PIC assembly on a structured mRNA in vitro. Ablating Ded1 interactions with eIF4A/eIF4E unveiled a requirement for the Ded1-CTD for robust initiation. Thus, Ded1 function in vivo is stimulated by independent interactions of its NTD with eIF4E and eIF4A, and its CTD with eIF4G.

*For correspondence:
ahinnebusch@nih.gov

## Introduction

Eukaryotic translation initiation is an intricate process that ensures accurate selection and decoding of the mRNA start codon. Initiation generally occurs through the scanning mechanism, which can be divided into the following discrete steps: (1a) 43S preinitiation complex (PIC) formation on the small (40S) ribosomal subunit by recruitment of methionyl initiator tRNA Met-tRNA$_i^{Met}$ in a ternary complex (TC) with GTP-bound eukaryotic initiation factor 2 (eIF2), facilitated by 40S-bound eIFs −1, −1A, −3, and −5; (1b) mRNA activation by binding of the eIF4F complex to the 7-methylguanosine capped 5' end; (2) 43S PIC attachment to the mRNA 5' end; (3) Scanning of the mRNA 5' untranslated region (UTR) by the PIC for a start codon in good sequence context; (4a) Start codon selection by Met-tRNA$_i^{Met}$ to form the 48S PIC; (4b) irreversible hydrolysis of GTP in the ternary complex, accompanied by release of eIF1 and its replacement on the 40S subunit by the eIF5 N-terminal domain; and (4 c) release of eIF2-GDP and eIF5 and recruitment of 60S subunit joining factor eIF5B by eIF1A; (5) Dissociation of eIF1A and eIF5B and joining of the 60S subunit to form the 80S initiation complex (*Jackson et al., 2010*; *Hinnebusch, 2014*).

The eIF4F complex, which stimulates 43S PIC recruitment to mRNA, comprises eIF4E, eIF4A, and eIF4G. The eIF4E (encoded by the *CDC33* gene in yeast) is a 24 kDa protein that binds directly to the 5' cap of the mRNA. eIF4A (encoded by *TIF1* and *TIF2* genes in yeast) is a 44 kDa DEAD-box RNA helicase thought to resolve mRNA structures that impede PIC attachment or scanning. eIF4G1 (encoded by *TIF4631* in yeast) is a 107 kDa scaffold protein harboring binding sites for RNA (named RNA1, RNA2, RNA3), the two other eIF4F components (eIF4E and eIF4A), and the poly(A) binding protein (PABP), hence promoting formation of a circular 'closed-loop' messenger ribonucleoprotein (mRNP). eIF4G can also interact with eIF3 (in mammals) or eIF5 (in yeast) to facilitate 43S PIC recruitment to the mRNA. eIF4G1 has a paralog, eIF4G2 (encoded by

*TIF4632* in yeast), which can make similar contacts with RNA and initiation factors and thereby promote initiation (*Clarkson et al., 2010*). The functions of these canonical eIF4F components have been studied in considerable detail (*Jackson et al., 2010*; *Hinnebusch, 2014*).

Recently, other DEAD-box RNA helicases besides eIF4A have been implicated in PIC attachment and scanning, including yeast Ded1 (homologous to Ddx3 in humans). Ded1 is an essential protein that stimulates bulk translation in vivo (*Chuang et al., 1997*; *de la Cruz et al., 1997*), and is especially important for translation of a large subset of yeast mRNAs characterized by long, structured 5' UTRs. Many such Ded1-hyperdependent mRNAs, identified by 80S ribosome footprint profiling of *ded1* mutants (*Sen et al., 2015*), were shown recently to require Ded1 in vivo for efficient 43S PIC attachment or subsequent scanning of the 5'UTR using the technique of 40S subunit profiling (*Sen et al., 2019*). Employing a fully reconstituted yeast translation initiation system, we further showed that Ded1 stimulates the rate of 48S PIC assembly on all mRNAs tested, but confers greater stimulation of Ded1-hyperdependent versus Ded1-hypodependent mRNAs (as defined by 80S ribosome profiling) in a manner dictated by stable stem-loop secondary structures in the 5'UTRs of the hyperdependent group (*Gupta et al., 2018*). Ded1 cooperates with its paralog Dbp1 in stimulating translation of a large group of mRNAs in vivo, and Dbp1 functions similarly to Ded1 in stimulating 48S PIC assembly in the yeast reconstituted system (*Sen et al., 2019*).

In addition to its canonical DEAD box helicase region comprised of two RecA-like domains, Ded1 contains additional N-terminal and C-terminal domains (NTD, CTD) (*Figure 1A*) that are not well conserved in amino acid sequence even within the subfamily comprised of Ded1 and mammalian Ddx3 helicases; and are thought to be largely unstructured (*Sharma and Jankowsky, 2014*). Distinct N-terminal and C-terminal extensions found immediately flanking the helicase core (NTE, CTE in *Figure 1A*) are relatively more conserved in the Ded1/Ddx3 subfamily, and at least for Ddx3, have partially defined structures and enhance the unwinding activity of the helicase core in vitro (*Floor et al., 2016*). Ded1 can interact in vitro with all three subunits of eIF4G, binding to the C-terminal RNA3 domain of eIF4G via the CTD, and interacting with eIF4A via the NTD (*Hilliker et al., 2011*; *Senissar et al., 2014*; *Gao et al., 2016*). Interaction of eIF4A with the Ded1-NTD stimulates the ability of purified Ded1 to unwind model RNA duplexes, whereas Ded1 interaction with eIF4G decreases the rate of RNA unwinding while increasing Ded1 affinity for RNA in vitro (*Gao et al., 2016*). Recently, we showed that the ability of Ded1 to stimulate 48S PIC formation in the reconstituted yeast system is impaired by elimination of either the Ded1-NTD, the Ded1-CTD, or the RNA2 or RNA3 domains of eIF4G1, with correlated defects for several mRNAs on removing the Ded1-CTD or eIF4G1 RNA3 domains that mediate Ded1/eIF4G1 interaction. These findings provided functional evidence that Ded1 association with the eIF4F subunits eIF4G and either eIF4A or eIF4E enhances Ded1 stimulation of 48S PIC assembly on native mRNAs (*Gupta et al., 2018*). It has been proposed that the majority of Ded1 exists in a stoichiometric complex with eIF4F in yeast cells (*Gao et al., 2016*).

There is evidence that the Ded1-CTD/eIF4G interaction promotes Ded1 stimulation of translation in cell extracts, and also the ability of Ded1, when highly overexpressed in cells, to repress translation and promote formation of P-bodies—cytoplasmic granules that function in storage or decay of translationally silenced mRNA. However, the in vivo importance of the Ded1 CTD in stimulating translation at native levels of Ded1 expression is unclear, as its elimination does not affect cell growth on nutrient-replete medium at low temperatures where *ded1* mutations generally have the strongest phenotypes (*Hilliker et al., 2011*). Instead, recent results indicate a role for the Ded1 CTD in down-regulating translation under conditions of reduced activity of the TORC1 protein kinase, which might entail increased degradation of eIF4G (*Aryanpur et al., 2019*). In contrast, the N-terminal region of Ded1 is clearly important for WT cell growth (*Hilliker et al., 2011*; *Floor et al., 2016*; *Gao et al., 2016*). There is also limited evidence that eIF4A binding to this region is functionally important in vivo, based on the finding that eIF4A overexpression can mitigate the growth phenotypes of a *ded1* allele with mutations in multiple domains, but not one only lacking the N-terminal 116 residues that encompass the eIF4A interaction site (*Gao et al., 2016*). However, this $\Delta$NTD$_{1-116}$ truncation might remove other functional determinants besides the eIF4A binding site that cannot be rescued by eIF4A overexpression, including a portion of the conserved NTE that enhances unwinding in vitro (*Floor et al., 2016*). Although Ded1 contains an 8-amino acid segment with similarity to a sequence in mammalian Ddx3 shown to bind eIF4E (*Shih et al., 2008*), and mutations in this Ded1 segment confer slow-growth phenotypes in yeast (*Hilliker et al., 2011*; *Senissar et al.,*

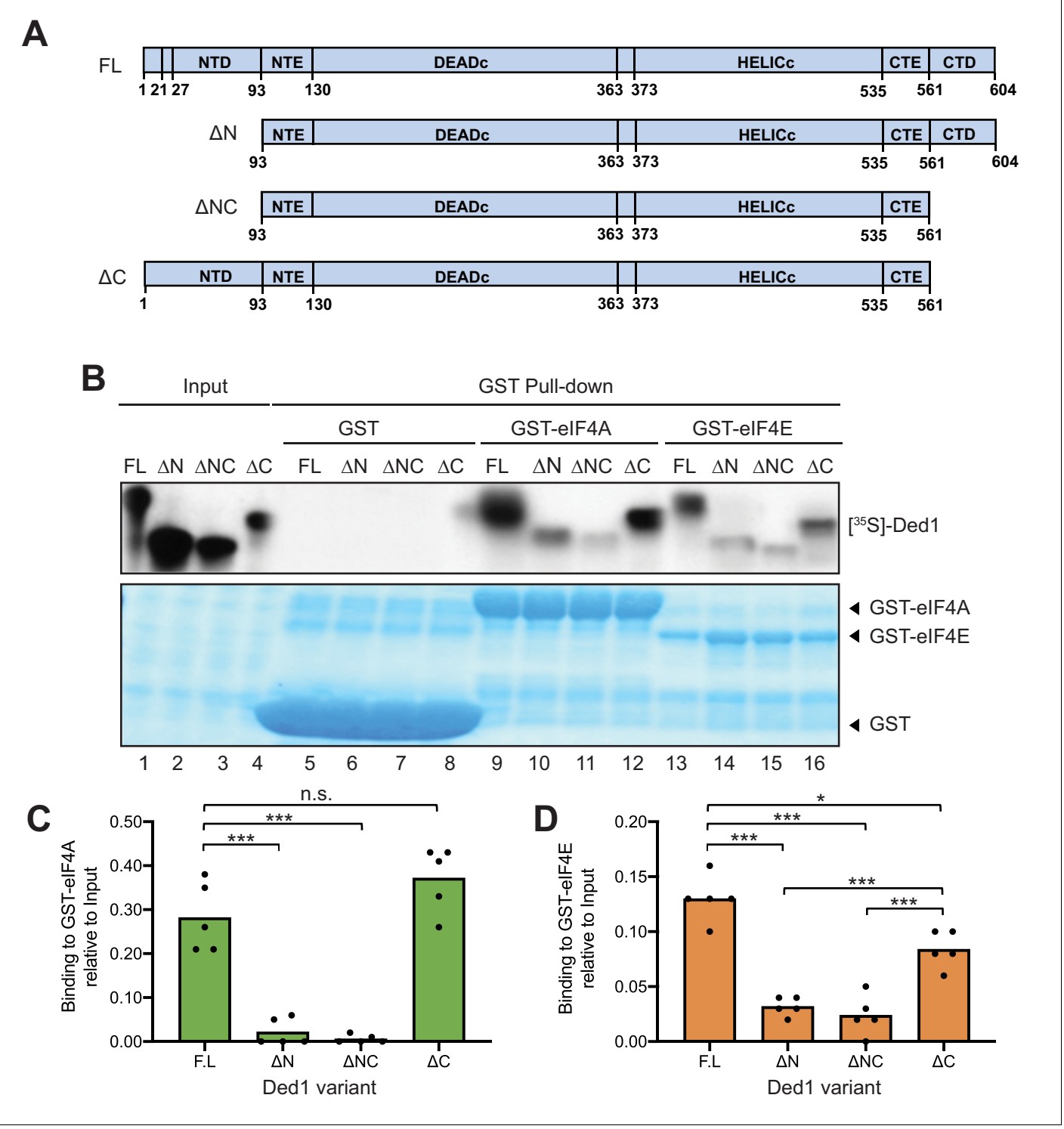

**Figure 1.** eIF4A and eIF4E interact primarily with the Ded1 N-terminus in vitro. (**A**) Schema of in vitro synthesized Ded1 variants, either full length (FL), lacking either the N-terminal domain (NTD) residues 2–92 (ΔN), C-terminal domain (CTD) residues 562–604 (ΔC), or both (ΔNΔC), used in GST pull-down assays. All derivatives contain the entire N- (NTE) and C-terminal (CTE) extensions and the two RecA domains (DEAD$_C$ and HELIC$_C$) responsible for RNA helicase activity. (**B**) The Ded1 NTD is required for strong binding to GST-tagged eIF4A and eIF4E. The amounts of [$^{35}$S]-labeled FL or truncated Ded1 proteins, visualized by fluorography (*upper panel*), present in the reactions (Input) or pulled down by GST, GST-eIF4A, or GST-eIF4A (visualized by Coomassie Blue staining, *lower panel*). (**C–D**) Quantification of the binding reactions for GST-eIF4A (**C**) or GST-eIF4E (**D**) by ImageJ analysis of fluorograms as in (**B**) from five replicate pull-down assays, expressing the amounts detected in the pull-downs as the percentages of input amounts. Individual dots show results of the replicates and bar heights give the mean values. n.s: not significant; *: p<0.05; **: p<0.01; ***: p<0.001.

*Figure 1 continued on next page*

Figure 1 continued

The online version of this article includes the following figure supplement(s) for figure 1:

**Figure supplement 1.** eIF4G interacts primarily with the Ded1 CTD in vitro.

2014), it was not shown that eIF4E binds to the Ded1 NTD nor requires the conserved motif for this interaction. As the binding determinants for eIF4E in Ded1 have not been identified, it is also unclear whether eIF4E interaction with Ded1 is physiologically important.

In this study, we set out to pinpoint the binding sites for eIF4A and eIF4E in the Ded1 NTD, which were not currently known, and to establish the importance of each individual interaction in promoting Ded1's ability to stimulate translation of Ded1-hyperdependent mRNAs in vivo. We have mutationally dissected Ded1 N-terminal residues extending up to the conserved NTE and identified specific amino acids whose substitution impaired Ded1 binding to recombinant eIF4A or eIF4E in vitro, which delineated non-overlapping NTD residues required for interaction with each factor. We could then demonstrate that NTD substitutions that selectively disrupt its association with eIF4A or eIF4E in vitro also impair association of Ded1 with native eIF4A or eIF4E in vivo, and diminish Ded1's ability to promote cell growth, bulk translation initiation, and translation of Ded1-dependent reporter mRNAs harboring defined stem-loop (SL) structures in their 5'UTRs, in yeast cells. We further showed that selectively disrupting Ded1 interactions with eIF4E or eIF4A by these NTD substitutions impaired Ded1 acceleration of 48S PIC assembly on a SL-containing reporter mRNA in the yeast reconstituted system. We provided additional genetic evidence that the key Ded1 NTD residues promote translation in vivo specifically by mediating interaction with eIF4A or eIF4E, and that this constitutes the critical in vivo function of the entire Ded1 NTD. Finally, we found that the Ded1 CTD becomes crucial for robust cell growth and translation initiation in vivo when Ded1's contacts with eIF4A or eIF4E are absent and its association with eIF4F must rely on Ded1-CTD interactions with eIF4G. Together, our results establish that the individual interactions of Ded1 with eIF4A, eIF4E and eIF4G that stabilize the eIF4F·Ded1 complex all promote Ded1 function in stimulating translation initiation in nutrient-replete yeast cells.

## Results

### Evidence that eIF4A and eIF4E both interact with the Ded1-NTD in vitro

Discrete binding determinants for eIF4A and eIF4E within the yeast Ded1 protein were unknown, which has made it difficult to assess the physiological importance of each interaction in vivo. To identify amino acids important for each interaction, we assayed binding of bacterially expressed glutathione-S-transferase (GST) fusions made to full-length yeast eIF4A1 or eIF4E to in vitro translated [$^{35}$S]-labeled Ded1 polypeptides, beginning with full-length (FL) Ded1 (amino acids 1–604) and truncated Ded1 variants lacking NTD residues 2–92 ($\Delta N_{2\text{-}92}$), CTD residues 562–604 ($\Delta C_{562\text{-}604}$), or both domains ($\Delta NC$) (*Figure 1A*). The labeled Ded1 polypeptides recovered with GST-eIF4A, GST-eIF4E, or GST alone on glutathione-agarose beads were resolved by SDS-PAGE, with the result that the FL Ded1 polypeptide bound to both GST-eIF4A and GST-eIF4E, but not GST alone (*Figure 1B*, lanes 9 and 13 vs. 5), confirming specific, separate interactions of Ded1 with both eIF4A and eIF4E. These and all subsequent pull-down assays included RNAase treatment to insure that the interactions were not bridged by RNA.

The amounts of bound Ded1 polypeptides were quantified and normalized to the input amounts for each reaction, and mean percentages of the input amounts that bound to GST-eIF4A or GST-eIF4E in replicate pull-down experiments were compared between FL Ded1 and the three truncated Ded1 variants. For GST-eIF4A, binding of the $\Delta C_{562\text{-}604}$ and FL Ded1 polypeptides were comparable, whereas drastically reduced amounts of the $\Delta N_{2\text{-}92}$ and $\Delta NC$ variants were bound (*Figure 1C*). These results support previous findings that eIF4A interacts specifically with the Ded1 NTD (*Gao et al., 2016*). Eliminating the NTD reduced Ded1 binding to GST-eIF4E as well, although binding was also reduced to a lesser extent by deleting the Ded1 CTD (*Figure 1D*). Importantly, however, comparing the results for $\Delta NC$ and $\Delta N_{2\text{-}92}$ revealed that the low-level binding conferred by $\Delta N_{2\text{-}92}$ was not significantly diminished by $\Delta C_{562\text{-}604}$, whereas $\Delta N_{2\text{-}92}$ exacerbated the moderate binding defect of

$\Delta C_{562\text{-}604}$ (*Figure 1D*). These results suggest that the Ded1 NTD is considerably more important than the CTD for eIF4E binding, and provide the first evidence that eIF4E binds directly to the yeast Ded1 NTD.

Ded1 has also been reported to interact through its C-terminus with eIF4G (*Gao et al., 2016*; *Hilliker et al., 2011*; *Senissar et al., 2014*). We verified this interaction under the conditions of our binding assay by determining that the $\Delta C_{562\text{-}604}$ truncation was sufficient to abolish binding by [$^{35}$S]-labeled Ded1 to a GST-eIF4G fusion expressed in bacteria (*Figure 1—figure supplement 1*, lanes 10 and 12). Thus, the Ded1 CTD is essential for binding to eIF4G but is either dispensable or relatively unimportant for interactions with eIF4A or eIF4E, respectively.

## eIF4A and eIF4E binding determinants in the Ded1 NTD are distinct and non-overlapping

To identify specific Ded1 residues critical for binding eIF4A and eIF4E in vitro, we generated [$^{35}$S]-labeled variants of the FL Ded1 polypeptide containing clustered alanine substitutions, or in one case a deletion, of ten blocks of conserved amino acids located throughout the NTD (*Figure 2A*). While the main consideration for sequence conservation was similarity among other *Saccharomyce-taceae* species (*Figure 2—figure supplement 1A*), most of these clusters include residues also conserved in higher eukaryotes (*Figure 2—figure supplement 1B*). Using the same GST pull-down assay as above, we observed that substituting Ded1 NTD residues 21–27, 29–35 and 51–57 conferred the greatest reductions in binding to GST-eIF4A (*Figure 2B*, cols. 5, 6, 8, bars in green hues), whereas residues 59–65 and 83–89 appear to be most important for binding to GST-eIF4E (*Figure 2C*, cols. 9,11, orange hues). These findings suggested that the critical binding determinants for eIF4A and eIF4E are located in adjacent, non-overlapping segments of the Ded1 NTD, located within residues 21–57 for eIF4A and within residues 59–89 for eIF4E.

To determine whether the non-contiguous binding determinants within each of these two intervals make additive contributions to binding eIF4A or eIF4E, we examined additional variants harboring combined substitutions of two different clusters. Whereas combining the contiguous substitutions 21–27 and 29–35 produced no further reduction in binding to GST-eIF4A compared to the single substitutions (*Figure 2B*, cols. 13 vs. 5 and 6), an additive reduction in binding was observed on combining the non-contiguous substitutions 21–27 and 51–57 (*Figure 2B*, cols. 14 vs. 5 and eight and *Figure 2—figure supplement 2*, cols. 5 vs. 1 and 2), which was comparable in its effect to deleting the entire ($\Delta NTD_{2\text{-}92}$) on binding to GST-eIF4A (*Figure 2B*, cols. 2 and 14, dark green hues). In contrast, combining the non-contiguous substitutions that individually impaired eIF4E binding produced reductions in binding by the 59-65/83-89 variant indistinguishable from those generated by the single substitutions (*Figure 2C*, cols. 16 vs. 9 and 11 and *Figure 2—figure supplement 2*, cols. 12 vs. 9 and 10). It is noteworthy that the two double substitutions that impair binding to GST-eIF4A (21-27/29-35 and 21-27/51-57) had little or no effect on binding to GST-eIF4E (*Figure 2B–C*, cols. 13–14), and that the double substitution 59-65/83-89 strongly reduced binding to GST-eIF4E with no significant effect on binding to GST-eIF4A (*Figure 2B–C*, col. 16). These last findings support the notion that the binding determinants for eIF4A and eIF4E are segregated into non-overlapping adjacent segments of the NTD.

Combining substitution 51–57 that selectively impairs binding to GST-eIF4A with the adjacent substitution 59–65 that selectively impairs binding to GST-eIF4E produced a binding defect to GST-eIF4E indistinguishable from that given by 59–65 alone (*Figure 2C*, cols. 15 vs. 8 and 9), supporting our conclusion that the binding determinants for eIF4E do not extend upstream into the region that binds eIF4A. However, the 59–65 substitution appeared to suppress the eIF4A binding defect of the adjacent upstream substitution 51–57 in the 51-57/59-65 variant (*Figure 2B*, cols. 15 vs. 8 and 9). One way to explain this 'context dependence' of the 51–57 substitution would be to propose that the residues in segment 59–65 that mediate eIF4E binding also antagonize binding of eIF4A to the contiguous segment 51–57 segment, such that their removal lessens the requirement for the eIF4A binding determinants in the 51–57 segment. Consistent with this possibility, we found that the 59–65 substitution did not suppress the stronger binding defect conferred by combining the non-contiguous substitution 21–27 with 51–57 in a 21-27/51-57/59-65 triple mutant (*Figure 2—figure supplement 3*, col. eight vs. col. 6). Regardless of the explanation, the 21-27/51-57 and 59-65/83-89 double substitutions provide Ded1 variants that selectively impair binding to eIF4A or eIF4E for our subsequent in vivo analysis.

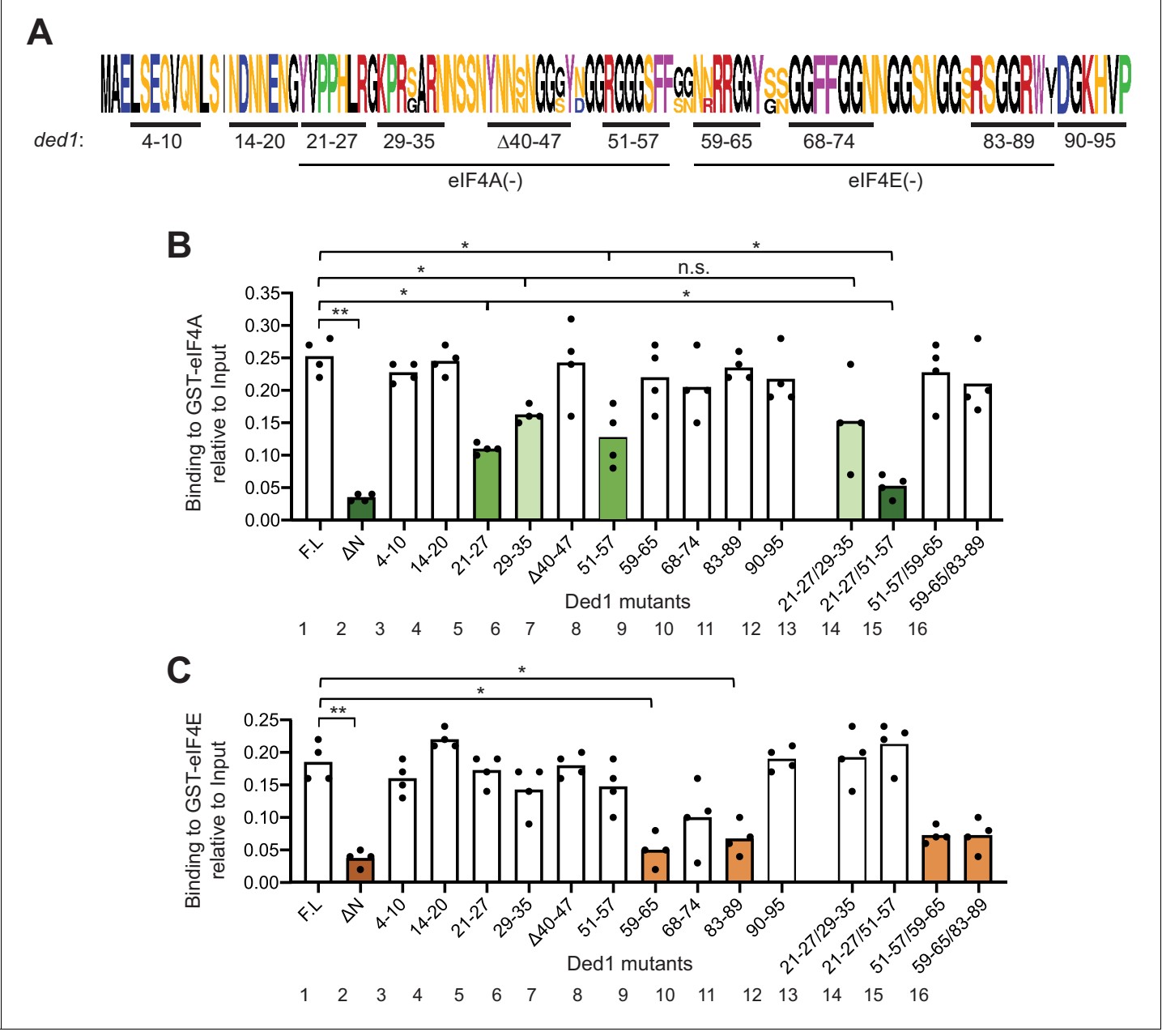

**Figure 2.** GST-eIF4A and GST-eIF4E bind to distinct non-overlapping segments of the Ded1 NTD. (A) WebLogo of amino acid sequence conservation in the Ded1 NTD among *Saccharomyces* species *S. cerevisiae, S. arboricola, S. kudriavzevii, S. bayanus, S. boulardii, S. mikatae, S. paradoxus,* and *S. pastorianus*. The blocks of residues chosen for clustered alanine substitutions, or in one case deletion (Δ40), of every residue in the block are underlined and labeled by the residue positions. The locations of segments implicated in binding to eIF4A or eIF4E by results in (B–C) are indicated. (B) Multiple segments in the N-terminal portion of the Ded1-NTD promote binding to GST-eIF4A. Results of pull-down assays using GST-eIF4A and the indicated FL or mutant Ded1 proteins, determined as in *Figure 1B–C* except using four replicates. Differences between mean values were analyzed with an unpaired students's t-test. n.s.: not significant, *: p<0.05, **: p<0.01. Shades of green indicate statistically significant decreases in mean values versus the FL construct, with darker shades indicating greater defects. (C) Multiple segments in the C-terminal portion of the Ded1-NTD promote binding to GST-eIF4E. Results of pull-down assays using GST-eIF4E and the indicated FL or mutant Ded1 proteins, determined as in *Figure 1B–C* except using four replicates. Shades of orange indicate statistically significant decreases in mean values versus the FL construct.
The online version of this article includes the following figure supplement(s) for figure 2:

**Figure supplement 1.** Multiple sequence alignment of the NTDs of Ded1/Ddx3 homologs and residues substituted here in the NTD of *S. cerevisiae* Ded1.

**Figure supplement 2.** Representative GST pull-down assays of key Ded1 NTD substitution mutants.

*Figure 2 continued on next page*

*Figure 2 continued*

**Figure supplement 3.** Evidence that substitution 59–65 rescues binding of the Ded1 51–57 variant, but not that of the 21-27/51-57 double mutant, to eIF4A.

**Figure supplement 4.** Highly conserved Ded1 residues Arg-27 and Trp-88 and moderately conserved Tyr-65 are all critical for Ded1 binding in vitro to GST-eIF4A or GST-eIF4E, respectively.

The Ded1 NTD is highly unstructured (*Floor et al., 2016*) and harbors RGG/RG motifs (*Rajyaguru and Parker, 2012*). To examine whether these motifs contribute to Ded1 interactions with eIF4A or eIF4E, we examined substitutions R27A, G28A, and R51A within the presumptive eIF4A-binding region identified above. Interestingly, R27A impaired binding to GST-eIF4A to the same extent observed on substituting the entire cluster 21–27 in which R27 resides, whereas G28A and R51A had no effect (*Figure 2—figure supplement 4B*). This finding is in accordance with the fact that R27 is highly conserved in the family *Saccharomycetaceae,* as well as in animals (*Figure 2—figure supplement 1A–B*), whereas, G28 and R51 are not. In contrast, despite its strong sequence conservation, Ala substitution of R62 within the region implicated in eIF4E binding had no significant effect on binding to GST-eIF4E (*Figure 2—figure supplement 4B*, *right* graph); although we note that the adjacent residue at position 61 is also an Arg in *S. cerevisiae.*

We further reasoned that because the Ded1 NTD is intrinsically unstructured, its hydrophobic and/or aromatic residues may be solvent-exposed and available for protein-protein interactions. Accordingly, we examined substitutions of five Phe or Tyr residues located within the NTD blocks of residues implicated above in binding to eIF4A or eIF4E. Strikingly, substituting each of the aromatic residues Y65 and W88 reduced binding to GST-eIF4E to the same extent observed for Ala substitutions of the entire corresponding segments 59–65 and 83–89, respectively (*Figure 2—figure supplement 4C*). By contrast, the Y21A and F56A/F57A substitutions within the eIF4A binding determinants 21–27 and 51/57, respectively, had no significant effect on binding to GST-eIF4A (*Figure 2—figure supplement 4C*). Together, these last findings reinforce the identification of segments 59–65 and 83–89 as binding determinants for eIF4E, and further suggest that aromatic residues Y65 and W88 within these segments might mediate key hydrophobic interactions with eIF4E. Although Y65 is not highly conserved, W88 is invariant among the eukaryotic species we examined (*Figure 2—figure supplement 1A–B*).

## Disruption of discrete binding determinants for eIF4A or eIF4E in the Ded1 NTD confers growth defects in vivo

Having identified substitutions in the Ded1 NTD that selectively reduce Ded1 binding to GST-eIF4A or GST-eIF4E in vitro, we addressed next whether these substitutions reduce Ded1 function in vivo. We began by asking whether the Ded1 NTD substitutions confer synthetic growth defects when combined with the temperature-sensitive (Ts⁻) *ded1-952* mutation, which strongly impairs growth at 37°C (*Sen et al., 2015*) but confers only a moderate slow-growth (Slg⁻) phenotype at 34°C (*Figure 3B*, rows 1–2) when introduced as a plasmid-borne Myc$_{13}$-tagged allele expressed from the native promoter (henceforth *ded1-ts*, *Figure 3A*) in a strain deleted for chromosomal *DED1*. Compared to *ded1-ts* alone, the *ded1-ts* alleles also containing the *21–27* or *51–57* mutations shown above to impair eIF4A binding in vitro conferred stronger Slg⁻ phenotypes at 34°C, with a slightly greater defect when the two mutations were combined within *ded1-ts-21-27/51-57* (*Figure 3B*, 34°C, rows 6–9). Similar findings were made for the *59–65, 83–89,* and *59-65/83-89* mutations that selectively impair eIF4E binding to Ded1 in vitro (*Figure 3B*, row 6 vs. 10–12), except that the reductions in growth were less pronounced than observed for the *21–27* and *21-27/51-57* mutations that impair eIF4A binding (*Figure 3B*, cf. rows 10–12 vs. 7 and 9). In contrast, the *14–20* and *Δ40–47* mutations that did not affect Ded1 binding to eIF4A or eIF4E in vitro also conferred no Slg⁻ in combination with *ded1-ts* in vivo. (The apparent suppression of the *ded1-ts* phenotype by *14–20* was not reproduced in independent transformants.) Western analysis showed that neither the double substitution *21-27/51-57* nor the quadruple substitution *21-27/51-57,59-65/83-89* reduced the steady-state level of the *ded1-ts* product (*Figure 3D*, lanes 5–7), indicating that they impair Ded1 function and not its expression.

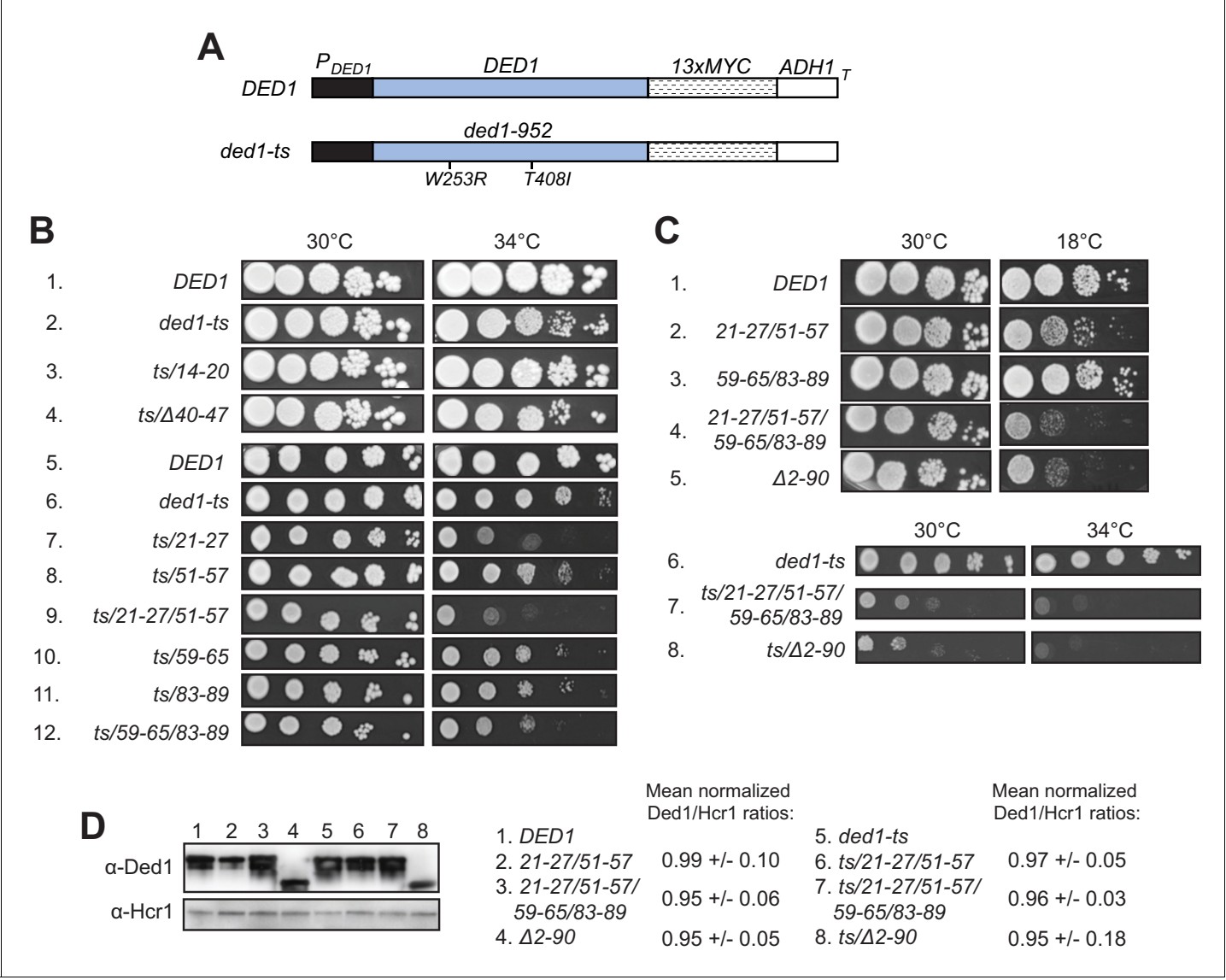

**Figure 3.** Clustered substitutions of Ded1 NTD residues that impair interaction with eIF4A or eIF4E in vitro confer growth defects in yeast. (A) Schematics of the myc₁₃-tagged parental *DED1* and *ded1-ts* alleles, expressed from the native *DED1* promoter ($P_{DED1}$) on a single copy (sc) plasmid, used to introduce NTD mutations described in *Figure 2A*. (B) Mutations substituting single or double binding determinants for eIF4A (*21-27; 51-57; 21-27/51-57*) or eIF4E (*59-65; 83-89; 59-65/83-89*) display synthetic temperature sensitivities with the *ded1-ts* allele of differing severity. Serial dilutions of yeast strains derived from yRP2799 by plasmid-shuffling containing the indicated derivatives of the *ded1-ts* (rows 2–12), or WT *DED1* allele (row 1), on sc *LEU2* plasmids (listed in *Table 3*) were spotted on synthetic complete medium lacking Leu (SC-Leu) and incubated at the indicated temperatures for 2-4d. (C) Disruption of eIF4A and eIF4E binding sites concurrently confers a severe growth defect comparable to that of NTD deletion *Δ2–90*. Yeast strains harboring the indicated derivatives of *DED1* (rows 1–5) or *ded1-ts* (rows 6–8) were analyzed as in (B). (D) Expression levels of the indicated mutants from (B) or (C) were assessed by Western analysis of WCEs extracted under denaturing conditions with TCA, using the indicated antibodies, following growth in SC-Leu at 18°C for *DED1* derivatives and 34°C for *ded1-ts* derivatives. Ded1/Hcr1 ratios of the indicated derivatives of the WT *DED1* or *ded1-ts* allele were obtained by ImageJ analysis of 3 independent experiments and are normalized to the corresponding parental allele's Ded1/Hcr1 ratios. This particular blot was atypical in suggesting a reduced level of the *ded1-ts-Δ2–90* product.

The online version of this article includes the following figure supplement(s) for figure 3:

**Figure supplement 1.** Substitutions of single Ded1 NTD residues that impair interaction with eIF4A or eIF4E in vitro confer growth defects in yeast.

The single amino acid substitutions Y21A and R27A in the eIF4A binding region each conferred a Slg⁻ phenotype at 34°C in combination with *ded1-ts,* which for *R27A* was greater than that observed for *Y21A* and comparable to that given by the *21–27* clustered substitution that encompasses both single-residue substitutions (*Figure 3—figure supplement 1*, rows 5–6 vs. 2–3). These phenotypes

correlate with the stronger eIF4A binding defect given by R27A versus Y21A, and with the similar binding defects observed for R27A and the 21–27 variant in vitro (*Figure 2—figure supplement 4B*). Although Y21A had little effect on Ded1 binding to eIF4A in vitro (*Figure 2—figure supplement 1*), the residue is highly conserved (*Figure 2—figure supplement 1*) and, hence, might have a greater impact in vivo. The F56A/F57A substitutions in the 51–57 interval, which had no effect on GST-eIF4A binding in vitro (*Figure 2—figure supplement 4B*) also had no effect on cell growth in combination with *ded1-ts* (*Figure 3—figure supplement 1*, rows 2 and 7). The Y65A and W88A single residue substitutions in the respective 59–65 and 83–89 segments of the eIF4E binding region each conferred moderate Slg⁻ phenotypes in combination with *ded1-ts* indistinguishable from those given by the corresponding *59–65* and *83–89* clustered substitutions (*Figure 3—figure supplement 1*, rows 11–12 vs. 9–10). These last findings are in accordance with the defects in GST-eIF4E binding in vitro shown above for these single and clustered substitutions (*Figure 2—figure supplement 4C*). Taken together, the effects of single-residue and clustered NTD substitutions on Ded1 binding to GST-eIF4A or GST-eIF4E in vitro are generally well correlated with their effects on cell growth in combination with the *ded1-ts* allele, with relatively stronger growth defects associated with mutations that reduce Ded1 binding to eIF4A versus those conferring comparable reductions in Ded1 binding to eIF4E.

We also examined the effects of the NTD mutations on Ded1 function in the absence of the *ded1-ts* mutation, finding that the *21-27/51-57* double cluster mutation affecting eIF4A binding confers a cold-sensitive Slg⁻ phenotype, which is exacerbated on combining it with the *59-65/83-89* double cluster mutations that impair eIF4E binding in vitro (*Figure 3C*, rows 1–4). Interestingly, the latter quadruple mutation conferred a strong Slg⁻ phenotype comparable to that given by the *ded1-Δ2–90* deletion allele lacking nearly the entire NTD, both in otherwise WT *DED1* (*Figure 3C*, rows 4–5) and in the *ded1-ts* allele, which suggests that binding to eIF4A and eIF4E constitutes the key in vivo function of the Ded1 NTD. Again, Western analysis revealed that the double and quadruple substitutions, as well as the Δ2–90 NTD deletion, had little effect on Ded1 steady-state expression levels (*Figure 3D*, lanes 1–4).

Finally, we asked whether disrupting binding of the Ded1 NTD to eIF4A or eIF4E would reveal an impact on cell growth of eliminating the Ded1 CTD and its known interaction with eIF4G. Consistent with previous findings (*Hilliker et al., 2011*), the CTD deletion that removes the C-terminal 43 residues of Ded1 (*ded1-ΔC*) confers no growth defect when the rest of Ded1 is intact. Importantly, however, the ΔC mutation exacerbates the cold-sensitive Slg⁻ phenotypes of all of the *ded1* mutations examined containing single- or double cluster substitutions in the NTD that impair binding to either eIF4A or eIF4E. While the exacerbation by ΔC is subtle for the *51–57, 59–65*, and *83–89* single-cluster mutations, it is pronounced for *21–27* and both double-cluster substitutions (*Figure 4A*, cf. adjacent rows). The exacerbation of the cold-sensitive growth phenotype of the NTD double-cluster mutations by ΔC occurred without reducing expression relative to the NTD variants with an intact CTD (*Figure 4B*, lanes 4–6). These findings are consistent with the idea that interaction of the Ded1 CTD with eIF4G is less critical for stabilizing the Ded1-eIF4E-eIF4A-eIF4G quaternary complex, activating Ded1 helicase function, or both, compared to the Ded1 NTD interactions with eIF4E and eIF4A; but that the importance of the CTD is increased when either the Ded1-eIF4E or Ded1-eIF4A interactions are impaired.

## Evidence that discrete binding determinants in the Ded1 NTD are important for interactions with eIF4A and eIF4E in vivo

Having identified eIF4A and eIF4E binding determinants in the Ded1 NTD in vitro and also demonstrating their contributions to Ded1 function in supporting cell growth, we sought next to demonstrate the importance of these interaction sites for Ded1 association with eIF4A or eIF4E in vivo using immunoprecipitation of myc-tagged Ded1 from yeast lysates. To this end, the $myc_{13}$-tagged *ded1-ts* alleles containing the *21-27/51-57* or *59-65/83-89* double-cluster mutations, *ded1-ts* containing no other mutations, or empty vector were introduced into a *DED1* strain lacking the chromosomal genes encoding eIF4G1 and eIF4G2 and expressing HA-tagged eIF4G1 from a plasmid under the eIF4G2 promoter, and whole cell extracts (WCEs) were immunoprecipitated with anti-Myc antibodies. The eIF4A, eIF4E, and eIF4G1-HA all coimmunoprecipitated specifically with the $myc_{13}$-tagged *ded1-ts* product, failing to be immunoprecipitated from the extract lacking a myc-tagged protein (*Figure 5*, rows 5–6), as expected for Ded1 interaction with eIF4F. The presence of

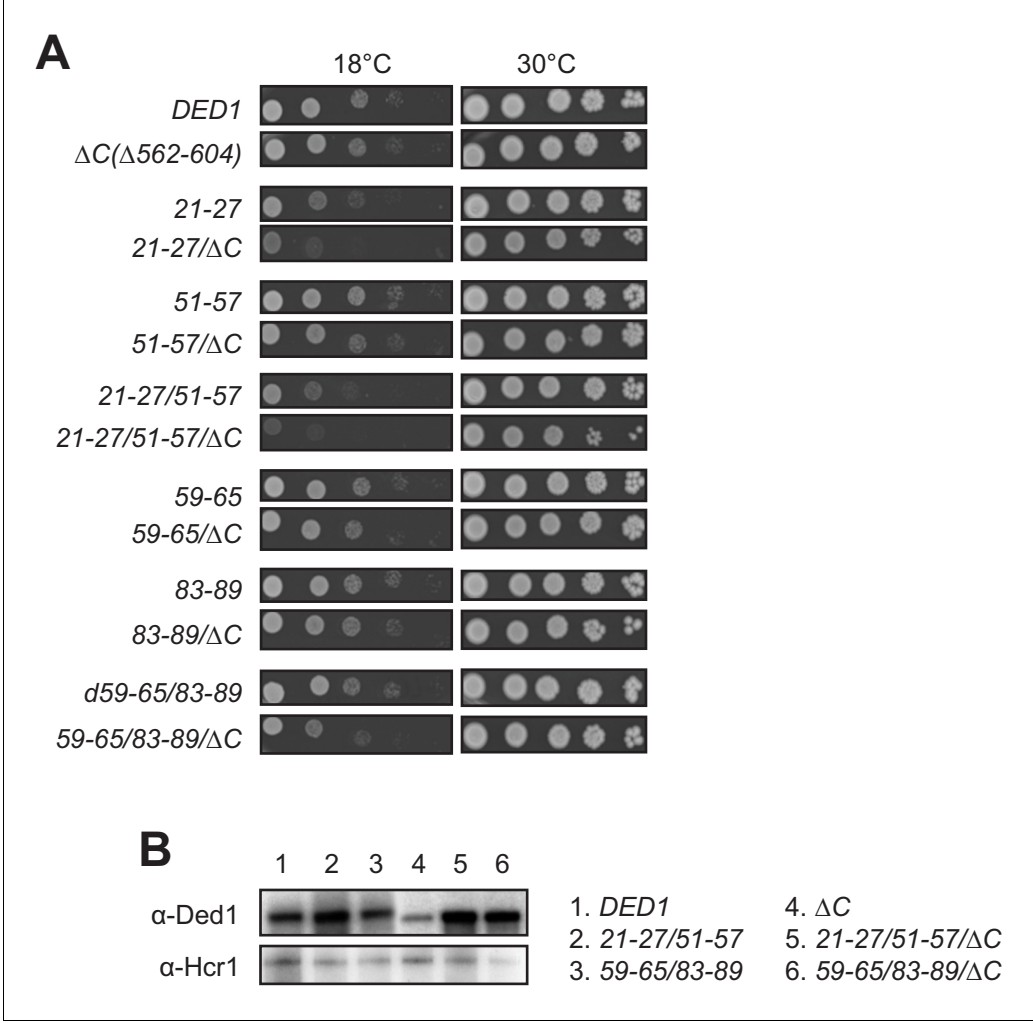

**Figure 4.** The Ded1 CTD is important for robust cell growth when NTD interactions with eIF4A or eIF4E are compromised. (**A**) Rates of colony formation of strains derived from yRP2799 containing the indicated derivatives of the WT *DED1* allele were analyzed as in *Figure 3B*. (**B**) Western analysis of the indicated mutants from (**A**) conducted as in *Figure 3D*.

mutations *21-27/51-57* in the *ded1-ts* allele, which impair Ded1 binding to GST-eIF4A in vitro (*Figure 2B*) reduced coimmunoprecipitation of the myc$_{13}$-tagged product with native eIF4A, but not eIF4E or eIF4G1-HA from WCEs (*Figure 5*, rows 6–7). Moreover, mutations *59-65/83-89*, which reduced Ded1 binding to GST-eIF4E in vitro (*Figure 2C*), specifically impaired coimmunoprecipition of eIF4E with the myc$_{13}$-tagged *ded1-ts/59-65/83-89* product (*Figure 5*, lanes 6 and 8). The results were unaffected by treating the immune complexes with RNAses A and T1 (*Figure 5*, lanes 9–12 vs. 5–8), suggesting that the interactions are not bridged by RNA. Thus, the binding determinants in the Ded1 NTD for eIF4A and eIF4E defined by in vitro pull-down assays with recombinant GST fusions to eIF4A or eIF4E are also crucial for association of native eIF4A or eIF4E with the *ded1-ts* product expressed at native levels in yeast cells. Because WT Ded1 is present in these cells, the reductions in eIF4E/eIF4A coimmunoprecipitation with the ded1-ts variants harboring NTD substitutions might have been intensified by competition with WT Ded1 for binding to eIF4F.

Our findings that the NTD substitutions selectively reduced association of the *ded1-ts* product with only eIF4E or eIF4A versus all three eIF4F subunits might indicate that Ded1 exists predominantly in binary complexes with each of the subunits of eIF4F; however, this would be odds with the expectation that the majority of Ded1 is associated with eIF4F in cells, based on estimates of the binding constant for Ded1 association with eIF4F and the cellular concentrations of these factors

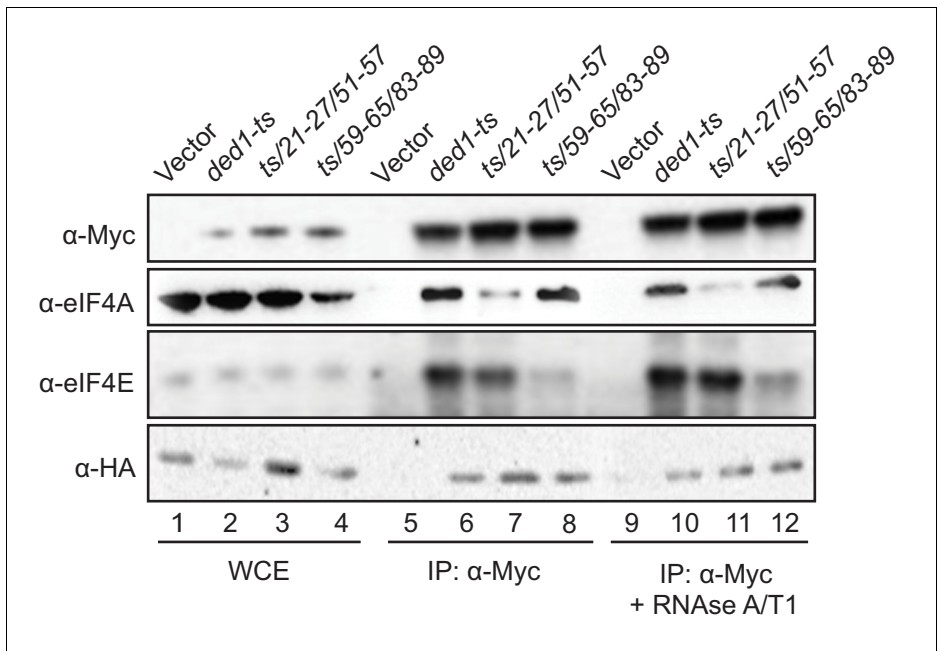

**Figure 5.** Disrupting Ded1 NTD binding determinants for eIF4A or eIF4E selectively impair association of the *ded1-ts* product with native eIF4A or eIF4E in yeast WCEs. Transformants of strain H4436 (expressing HA-tagged eIF4G1 and lacking eIF4G2) harboring the indicated derivatives of (myc$_{13}$-tagged) *ded1-ts* were cultured in SC-Leu-Trp medium and WCEs prepared under non-denaturing conditions (lanes 1–4) were immunoprecipitated with anti-myc antibodies (lanes 5–12). Aliquots corresponding to 5% of the WCEs and 50% of the resulting immune complexes, either without treatment (lanes 5–8) or following 30 min treatment with RNAse A/T1 at room temperature (lanes 9–12), were subjected to Western analysis with the indicated antibodies.

(*Gao et al., 2016*). Rather, it seems plausible that weakening the association of Ded1 with eIF4E leads to a specific reduction in eIF4E coimmunoprecipitation with ded1-ts owing to dissociation of eIF4E from the scaffold subunit eIF4G during the extensive washing steps involved in the experiment, notwithstanding the known stable interaction of eIF4E with eIF4G (*Mitchell et al., 2010*). The relatively lower affinity of yeast eIF4A for eIF4G (*Mitchell et al., 2010*; *Park et al., 2012*) makes this explanation even more likely for the selective loss of eIF4A association with mutated ded1-ts; although the existence of Ded1-eIF4A binary complexes in vivo has also been predicted (*Gao et al., 2016*).

## Genetic evidence that eliminating interactions with eIF4A or eIF4E is responsible for growth defects conferred by substituting Ded1-NTD binding determinants

We sought next to provide evidence that eliminating the binding determinants for eIF4A or eIF4E in the Ded1 NTD confer growth defects owing to loss of the specific interaction with eIF4A or eIF4E, respectively. We reasoned that if a Ded1 NTD substitution confers a growth defect, but can still partially interact with its binding partner, then overexpression of the partner should reinstate the interaction by mass action and rescue normal cell growth. If however the Ded1 mutant cannot bind to the partner at all, then overexpressing the latter might have little effect on cell growth (*Figure 6A*). To test this prediction, we overexpressed eIF4A1 from a high-copy (hc) *TIF1* plasmid in the panel of *ded1-ts* mutants with single or double cluster substitutions in the eIF4A binding region and measured growth rates by cell-spotting assays. eIF4A overexpression mitigated the Slg⁻ phenotypes at 34°C of the *21–27, 29–35*, and *51–57* single-cluster mutations, and also the *21-27/29-35* double cluster mutations (*Figure 6B*, rows 3–10, cf. adjacent rows), all of which conferred partial reductions in Ded1 binding to GST-eIF4A in vitro (*Figure 2B*), consistent with restored interaction of Ded1-eIF4A association by mass action. Importantly however, eIF4A overexpression did not mitigate the stronger Slg⁻ phenotype conferred by the *ded1-ts,21-27/51-57* double-cluster mutation (*Figure 6B*, rows 11–

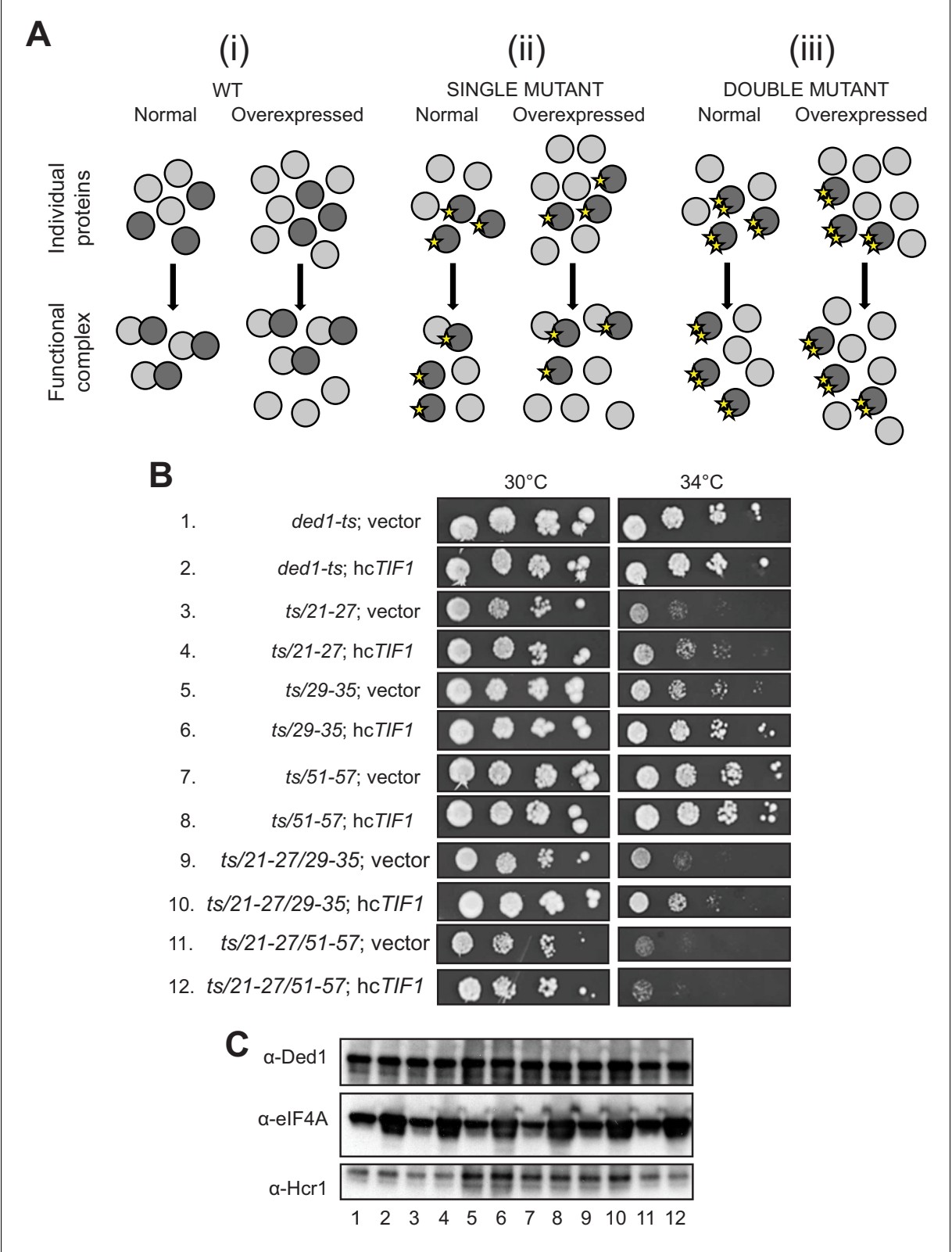

**Figure 6.** Evidence that Ded1-NTD binding determinants of eIF4A promote cell growth by enhancing eIF4A association. (**A**) Schema summarizing expected outcomes for Ded1-eIF4A association based on mass action on overexpressing eIF4A (grey circles) in cells containing different *ded1-ts* proteins (dark grey circles), as follows: (i) otherwise WT; (ii) a *ded1-ts* derivative lacking a single binding determinant (single star) that only reduces binding to eIF4A, or (iii) a *ded1-ts* derivative lacking two binding determinants (double star) that essentially abolishes eIF4A binding. Ded1-eIF4A

*Figure 6 continued on next page*

*Figure 6 continued*

association depicted by overlapping the circles. (B) Derivatives of strain yRP2799 containing the indicated *ded1-ts* alleles harboring hc*TIF1* plasmid pBAS3432 or empty vector were examined for rates of colony formation at the indicated temperatures as in *Figure 3B*. (C) Western blot analysis of the strains in (B) conducted as in *Figure 3D* using the indicated antibodies.

The online version of this article includes the following figure supplement(s) for figure 6:

**Figure supplement 1.** Derivatives of strain yRP2799 containing the indicated *ded1-ts* alleles harboring hc*TIF1* plasmid pBAS3432 or empty vector were examined for rates of colony formation at the indicated temperatures as in *Figure 6*.

12), which reduced binding to GST-eIF4A to the low level conferred by the $\Delta N_{2-92}$ deletion that removes the entire eIF4A binding domain (*Figure 2B*). Nor did eIF4A overexpression mitigate the Slg⁻ phenotype conferred by the *59-65/83-89* mutations that selectively impair binding to eIF4E (*Figure 6—figure supplement 1*), supporting the interpretation that suppression of the other NTD mutations that partially impair eIF4A binding results from restoration of Ded1-eIF4A association by mass action (*Figure 6A*). Western analysis verified that eIF4A was overexpressed similarly in all of the *ded1* mutants (*Figure 6C*). These findings support the idea that Ded1 substitutions in the eIF4A-binding region of the NTD confer growth defects in vivo owing to impaired association with eIF4A in cells.

When we subjected the *ded1-ts* alleles containing single- or double cluster mutations in the eIF4E binding region to a similar analysis by overexpressing eIF4E from a hc *CDC33* plasmid, we saw a uniform, modest exacerbation of growth defects for all strains (not shown), which may indicate that eIF4E overexpression is toxic for yeast. Instead, we exploited the fact that overexpressing WT *DED1* can mitigate the growth defects conferred by the chromosomal eIF4E mutant allele *cdc33-1* (*de la Cruz et al., 1997*). In agreement with this, we found that overexpressing either WT *DED1* or the *ded1-21-27/51-57* allele defective for eIF4A binding from a hc plasmid increased the growth rate of

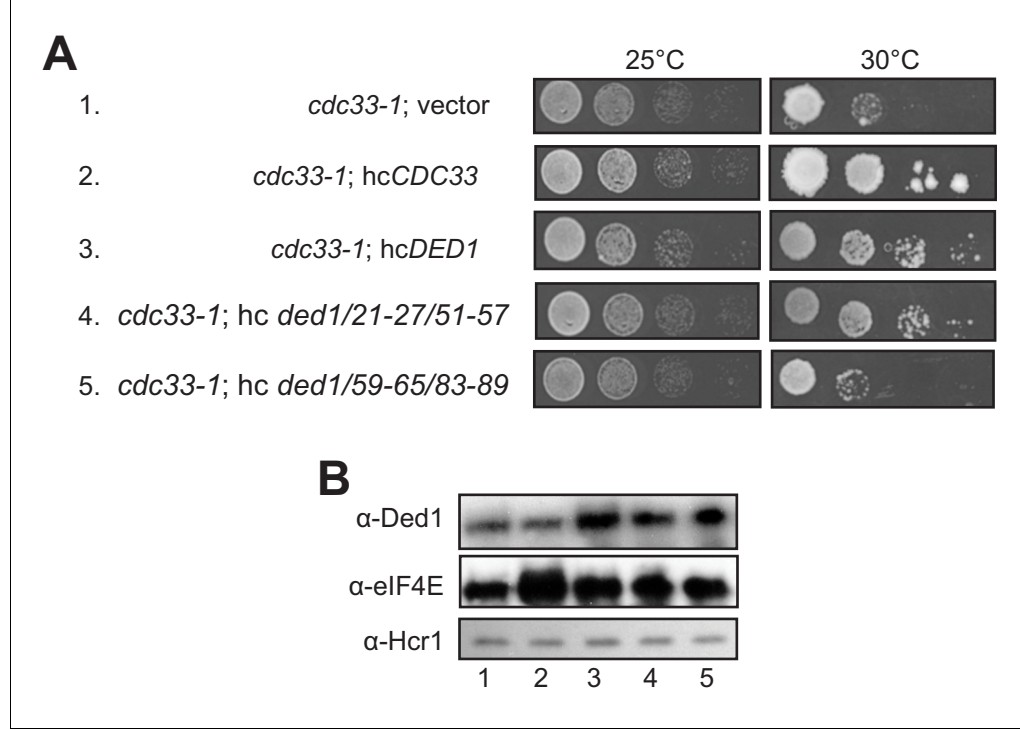

**Figure 7.** Evidence that Ded1-NTD binding determinants of eIF4E promote cell growth by enhancing eIF4E association. (A) Transformants of *cdc33-1* mutant F696 harboring empty vector YEplac195, hc*CDC33* plasmid (p3351), or hc plasmids with the indicated WT (p4504) or mutant (pSG48 or pSG49) *DED1* alleles were examined for rates of colony formation at the indicated temperatures as in *Figure 3B*. (B) Western blot analysis of the strains in (A) conducted as in *Figure 3D* using the indicated antibodies.

*cdc33-1* cells (*Figure 7A*, rows 3–4 vs. 1. Importantly, however, the hc *ded1-59-65/83-89* allele, defective for eIF4E binding in vitro, had no effect on growth of the mutant cells (*Figure 7A*, row 5 vs. 1). Western analysis verified that the mutant and WT *ded1* alleles were overexpressed similarly in the *cdc33-1* mutant (*Figure 7B*). These findings have two important implications. First, they provide strong evidence that the *ded1-59-65/83-89* mutation impairs both a physical and functional interaction between Ded1 and eIF4E in cells. Second, they imply that overexpressing Ded1 partially suppresses the *cdc33-1* mutation by enhancing interaction of the *cdc33-1* product with Ded1 by mass action, presumably increasing the concentration of the eIF4F·Ded1 complex in which Ded1 functions most efficiently, rather than indirectly compensating for a reduction in eIF4F function (that should occur for all *ded1* alleles).

## Disruption of Ded1-eIF4A or eIF4E interactions impairs translation in vivo

We next investigated changes in bulk translation initiation upon disruption of Ded1-eIF4A or Ded1-eIF4E interactions by Ded1 NTD mutations, examining first the effects on total polysome assembly. Cells were treated with cycloheximide just prior to harvesting to prevent polysome run-off during isolation, and polysomes were resolved from 80S monosomes, free 40S and 60S subunits, and free mRNPs by sedimentation through sucrose density gradients. Both the *ded1-ts,21-27/51-57* and *ded1-ts,59-65/83-89* mutations, impairing Ded1 interactions with eIF4A or eIF4E, respectively, reduced the ratio of polysomes to monosomes (P/M), indicating a reduction bulk translation initiation, which was not significantly greater for one mutation versus the other (*Figure 8A*).

We further analyzed the functional defects in eIF4A- and eIF4E binding mutants by measuring expression of three luciferase (*FLUC*) reporter mRNAs harboring a Ded1-hypodependent synthetic unstructured 5'UTR harboring no stem-loop (no SL), or either cap-distal or cap-proximal SLs shown previously to confer Ded1-hyperdepence both in vivo (*Sen et al., 2015*) and in vitro (*Gupta et al., 2018*; *Figure 8B*). Relative to the *ded1-ts* parental mutant, introducing single- or double-cluster NTD mutations that impair binding to eIF4A or eIF4E, or control NTD mutations *14–20* and *40–47* with no effect on eIF4A/eIF4E binding, had little or no effect on expression of the reporter lacking a 5'UTR SL (*Figure 8C*, dark gray bars). The reporter with a cap-distal SL showed reduced expression relative to the no-SL construct in the *ded1-ts* parental strain (*Figure 8—figure supplement 1*, cols. 1–2), which was unaffected by the control NTD mutations; but was driven even lower by all of the mutations affecting eIF4A or eIF4E binding, with the greatest reductions seen for the two double substitutions *ded1-ts,21-27/51-57* and *ded1-ts,59-65/83-89* that eliminate, respectively, eIF4A or eIF4E binding to Ded1 (*Figure 8C*, light gray bars). Expression of the cap-proximal SL reporter was also reduced somewhat in the *ded1-ts* parental strain (*Figure 8—figure supplement 1*, col. 3 vs. 1), and was not further diminished by the control NTD mutations or the single cluster mutants defective for eIF4E binding (*59–65* and *83–89*), whereas the single cluster mutants defective for eIF4A binding (*21–27* and *51–57*) and both double-cluster mutants impairing eIF4A or eIF4E binding showed substantially reduced expression of this reporter, with the greatest reductions seen for the two double-cluster mutants (*Figure 8C*, white bars). Interestingly, overexpression of eIF4A conferred increased expression of the no SL and cap-distal reporters in the *ded1-ts/21–27* mutant, predicted to exhibit partially impaired binding of eIF4A to Ded1, but not in the *ded1-ts/21-27/51-57* or *ded1-ts/59-65/83-89* strains expected to be fully defective for eIF4A binding, or for eIF4E binding, respectively (*Figure 8—figure supplement 5*). This supports the notion that the defect in reporter expression conferred by *ded1-ts/21–27* involved impaired Ded1 association with eIF4A.

Finally, we verified that the decreased expression of the cap-proximal *FLUC* reporter in the Ded1 mutants results from impaired translation by examining the effects of the mutations on the polysome size distributions of the reporter mRNA, assayed by qRT-PCR of total mRNA isolated from each fraction of the density gradient used to resolve total ribosomal species, noting that a shift in mRNA abundance from larger to small polysomes, or from polysomes/monosomes to free mRNPs, indicates a reduction in the rate of translation initiation. In the parental *ded1-ts* mutant, the reporter mRNA is almost equally distributed among all gradient fractions (*Figure 8D (ii)*, gray points). In the eIF4E-binding mutant *ded1-ts,59-65/83-89,* the proportion of reporter mRNA in the heavy polysome fractions is reduced, and the proportion in the smallest polysomes and monosomes is increased (*Figure 8D (ii)*, orange points) relative to the mRNA distribution in the parental strain (*Figure 8D (ii)*, grey points). Moreover, in the eIF4A binding mutant *ded1-ts,21-27/51-57*, the proportion of reporter

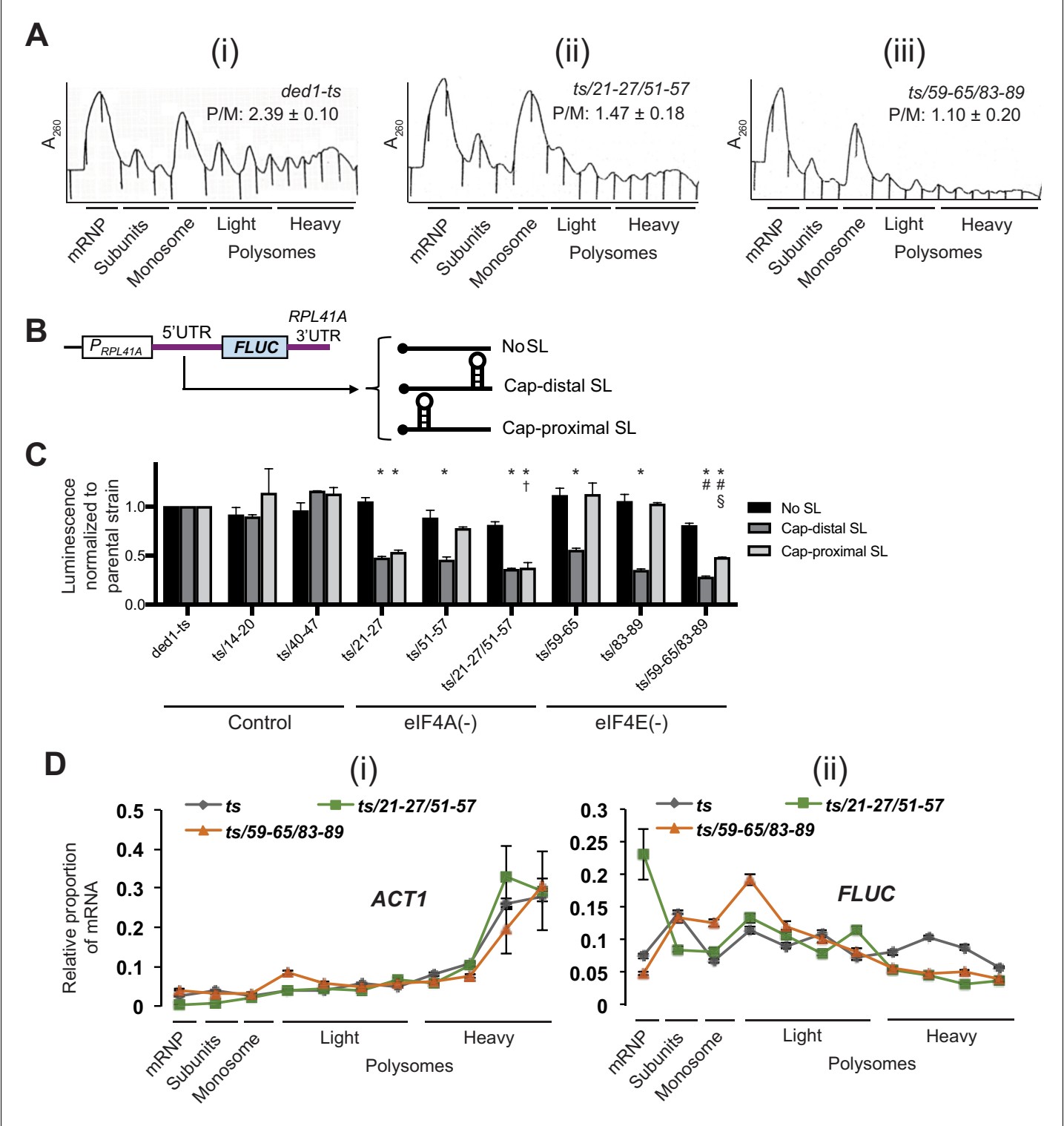

**Figure 8.** Disruption of Ded1 NTD interactions with eIF4A or eIF4E confer bulk and mRNA-specific translation defects in vivo. (**A**) Polysome profiling of derivatives of strain yRP2799 containing the indicated *ded1-ts* alleles lacking binding determinants for eIF4A (ii) or eIF4E (iii) exhibit a decrease in bulk polysome assembly compared to the parental *ded1-ts* strain (i). Strains were cultured in SC-Leu medium after shifting from 30°C to 36°C for 3 hr, and WCE extracts were resolved by velocity sedimentation and scanned at 260 nm. The mean ratios of polysomes to monosomes (P/M) determined from three replicate WCEs are indicated. The mean P/M ratios for each mutant differ significantly from the WT mean ratio (both p-values<0.002), but not from each other (p=0.08). Parallel analysis of isogenic *DED1+* cells revealed, as expected, a significantly greater P/M ratio (3.90 + / - 0.11) compared to that shown in the figure for *ded1-ts* (2.39 + / - 0.10). (**B**) Schema of the reporter constructs employed to interrogate Ded1-dependent translation in vivo,

*Figure 8 continued on next page*

*Figure 8 continued*

containing the *RPL41A* promoter and 3'UTR containing derivatives of the *RPL41A* 5'UTR harboring 23 tandem repeats of CAA nucleotides (nt) and designed to be largely unstructured (No SL, pFJZ342), or additionally containing a stem-loop insertion (of predicted ΔG of −3.7kcal/mol) located 55 nt from the 5'end (Cap-distal SL, pFJZ623), or a SL (of predicted ΔG of −8.1kcal/mol) at seven nt from the 5'end (Cap-proximal SL, pFJZ669). (C) Transformants of strains derived from yRP2799 with the indicated *ded1-ts* alleles and reporter constructs from (B) were cultured in SC-Leu at 34°C for two doublings and mean specific luciferase activities were determined from three independent transformants and normalized to that obtained for the reporters in the *ded1-ts* parental strain, which were set to unity.*: Significant differences in mean values compared to *ded1-ts*; †: compared to *ded1-ts/ 51–57*; #: compared to *ded1-ts/59–65*; §: compared to *ded1-ts/83–89*, as indicated by a p-value of <0.05 in an unpaired student's t-test. The effects of substitutions in impairing binding to eIF4A, eIF4E, or neither protein (Control), are indicated at the bottom. (D) Distributions of *ACT1* mRNA (i), or Cap-proximal SL reporter mRNA (ii), across sucrose gradients following velocity sedimentation of WCEs of strains from (C) containing the indicated *ded1* alleles and the Cap-proximal SL reporter. qRT-PCR was conducted on total RNA purified from each fraction using primers specific for *ACT1* or *FLUC* mRNA, and the abundance of each transcript was normalized to that of a set of 'spike-in' controls and plotted as a fraction of the total normalized abundance in the entire gradient. The mean values and S.E.M.s determined from three replicate gradients are plotted.

The online version of this article includes the following figure supplement(s) for figure 8:

**Figure supplement 1.** Substitutions of single Ded1 NTD residues that impair interaction with eIF4A or eIF4E in vitro confer translation initiation defects in yeast.

**Figure supplement 2.** Combining four Ded1-NTD clustered substitutions that individually impair interaction with eIF4A or eIF4E in vitro is comparable to deletion of the Ded1 NTD in reducing translation initiation vivo.

**Figure supplement 3.** Removal of the Ded1 CTD impairs translation initiation in vivo only when NTD interactions with either eIF4A or eIF4E are compromised.

**Figure supplement 4.** Data from *Figure 8D (ii)* for the cap-proximal SL *FLUC* mRNA was analyzed using a Student's unpaired t-test to compare the mean *FLUC* mRNA proportions calculated from three replicates for each polysome fraction between the *ts/21-27/51-57* and *ts/59-65/83-89* mutant strain and the parental strain (*ts*).

**Figure supplement 5.** *hcTIF1* mitigates the reduction in *LUC* reporter expression conferred by the *ts/21–27* mutation but not by the *ts/21-27/51-57* or *ts/59-65/83-89* mutations.

mRNA in heavy polysomes is reduced and the proportion in free mRNP is elevated (*Figure 8D (ii)*, green points) compared to that seen in the *ded1-ts* strain (*Figure 8D (ii)*, grey points). The more extensive shift from polysomes to free mRNP observed for the eIF4A-binding mutant compared to the shift from larger to smaller polysomes/monosomes conferred by the eIF4E-binding mutant (*Figure 8D (ii)*, green vs. orange) is consistent with the relatively greater reductions in cap-proximal SL reporter expression given by the set of three eIF4A-binding mutants versus the three eIF4E-binding mutants (*Figure 8C*, eIF4A(-) vs. eIF4E(-) mutants). This shift of the reporter mRNA from polysomes to free mRNP in the eIF4A-binding mutant or to smaller polysomes in the eIF4E-binding mutant is statistically significant (*Figure 8—figure supplement 4*). In contrast to the behavior of the reporter mRNA, the polysome distribution of native *ACT1* mRNA was not substantially altered by either of the Ded1 NTD mutations (*Figure 8D* (i)), suggesting that Ded1's interactions with eIF4A and eIF4E are relatively more important for the reporter mRNA harboring a stable SL structure compared to *ACT1* mRNA. *ACT1* mRNA was not found to be hyperdependent on Ded1 by ribosome profiling of a *ded1* mutant (*Sen et al., 2015*). These results confirm that loss of eIF4A and eIF4E interactions by the Ded1 NTD lead to reduced translation initiation of a Ded1-hyperdependent reporter mRNA, as well as to decreased bulk translation initiation in vivo.

Assaying expression of the same *LUC* reporters, we found that the single amino acid mutations in the eIF4A binding region *Y21A* and *R27A* reduced expression of only the cap-distal SL (*Y21A*) or both SL reporters (*R27A*), and that the *Y65A* and *W88A* single-residue mutations in the eIF4E binding region reduced expression of both SL reporters, compared to the parental *ded1-ts* strain (*Figure 8—figure supplement 1*), thus supporting the importance of these individual residues in Ded1 NTD interactions with eIF4A and eIF4E in vivo.

We also provided evidence that eliminating both eIF4A and eIF4E binding simultaneously essentially inactivates the Ded1 NTD in promoting bulk translation as well as translation of Ded1-hyperdependent reporter mRNAs. As shown in *Figure 8—figure supplement 2*, the *21-27/51-57/59-65/83-89* quadruple-cluster mutation and the *Δ2–90* deletion of the NTD introduced into otherwise WT *DED1* conferred nearly indistinguishable reductions in bulk polysome assembly (panel A) and expression of both cap-proximal SL and cap-distal SL *FLUC* reporters (panel B). In both assays, the defects were quantitatively smaller than shown above for the double-cluster mutations in *Figure 8A* and

*5C*, which we attribute to the absence of the *ded1-ts* mutation in the parental and mutant alleles being compared in the current assays.

Finally, we obtained evidence that the effect of NTD mutations in exacerbating the effect of deleting the Ded1 CTD shown above in cell growth assays (*Figure 4*) could also be observed in the functional assays for polysome assembly and reporter mRNA expression. As shown in *Figure 8—figure supplement 3A*, the *ded1-ΔC* mutation analyzed above has little or no effect on polysome assembly on its own (cf. panels (i)-(ii)). However, ΔC clearly exacerbates the effects of both the *21-27/51-57* and *59-65/83-89* double-cluster mutations in the Ded1 NTD in reducing P/M ratios, with a relatively greater effect for the *21-27/51-57* mutation that eliminates eIF4A binding (cf. (iii)-(iv) and (v)-(vi)). Similarly, *ded1-ΔC* alone has only small effects on expression of the SL-containing *FLUC* reporters, but ΔC exacerbates the reductions in reporter expression conferred by *21-27/51-57* (*Figure 8—figure supplement 3B*, set four vs. sets 2 and 3) and by the *59-65/83-89* mutation (*Figure 8—figure supplement 3B*, set six vs. sets 2 and 5). These findings support our conclusion that eliminating eIF4G association with the Ded1 CTD imposes a relatively greater requirement for the Ded1 NTD interactions with eIF4A and eIF4E, either in forming or stabilizing the eIF4F·Ded1 complex or in stimulating Ded1 unwinding activity.

## Disruption of Ded1-eIF4A or eIF4E interactions impairs 48S PIC assembly on a SL-containing mRNA in the purified system

The yeast reconstituted system provides a unique mechanistic view of translation initiation (*Acker et al., 2007*; *Algire et al., 2002*; *Gupta et al., 2018*; *Mitchell et al., 2010*), which allowed us previously to reconstitute the function of Ded1 in stimulating the rate of 48S PIC assembly on native mRNAs, and to demonstrate substantially greater rate-enhancement for mRNAs that harbor structured 5'UTRs that confer hyperdependence on Ded1 for efficient translation in vivo, compared to mRNAs with less structured 5'UTRs that are Ded1-hypodependent in cells. Moreover, we showed that deleting the N-terminal 116 residues of Ded1 ($\Delta NTD_{1-116}$) reduced the maximum rate of PIC assembly achieved at saturating levels of Ded1 ($k_{max}$) and increased the concentration of Ded1 required for the half-maximal rate ($K_{1/2}$) for several mRNAs containing stable SL structures in the 5'UTRs (*Gupta et al., 2018*). As eliminating the entire N-terminal region did not distinguish between impairing binding to eIF4A, eIF4E, or eliminating some other stimulatory function of this region of Ded1, we sought to determine whether selectively impairing Ded1 interaction with eIF4A or eIF4E would impair Ded1 acceleration of 48S assembly on a SL-containing mRNA. Accordingly, we purified full-length (FL) WT Ded1, Ded1-$\Delta NTD_{1-116}$ and the two Ded1 variants harboring the double clustered substitutions that disrupt binding to eIF4A 21-27/51-57, or the single cluster substitution 59–65 that reduces binding to eIF4E, and compared them for acceleration of 48S PIC assembly on an mRNA (dubbed *CP-8.1*) containing the same cap-proximal SL present in the *FLUC* reporter mRNA analyzed above appended to the coding sequences and 3'UTR of of the Ded1-hypodependent native *RPL41A* mRNA. The three purified mutant proteins had ATPase activities similar to that of WT Ded1 (data not shown).

In agreement with previous results (*Gupta et al., 2018*), FL Ded1 substantially increased the $k_{max}$ of 48S PIC formation on *CP-8.1* mRNA by ~5 fold compared to the absence of Ded1, and deleting the NTD diminished the rate enhancement conferred by addition of Ded1 compared to no Ded1 in the reaction by ≈60% (*Figure 9A*, cols. 1–3; see *Figure 9—figure supplement 1C* for data summary). Interestingly, disrupting eIF4A binding to the NTD with the 21-27/51-57 double substitutions was comparable to the $\Delta NTD_{1-116}$ in reducing $k_{max}$, whereas the 59–65 substitution that affects eIF4E binding had no effect on the $k_{max}$ (*Figure 9A*, cols. 3–5). The $\Delta NTD_{1-116}$ truncation also greatly increased the $K_{1/2}$ for Ded1 on this mRNA (*Figure 9B*, cols. 1–2), as observed previously (*Gupta et al., 2018*); and both the 21-27/51-57 and 59–65 substitutions also increased the $K_{1/2}$ for Ded1 to an extent ≈40% of that given by the $\Delta NTD_{1-116}$ (*Figure 9B*, cols. 2–4). These findings suggest that binding of eIF4A, but not eIF4E, to the Ded1 NTD is required for maximum acceleration of 48S PIC assembly on *CP-8.1* mRNA, but that both eIF4A and eIF4E interactions decrease the amount of Ded1 required to achieve this stimulation, possibly by enhancing formation of the eIF4F·Ded1 complex (*Gupta et al., 2018*).

We showed previously that Ded1 also increases the rate of 48S PIC assembly on Ded1-hypodependent *RPL41A* mRNA, but to a lesser degree and at a much lower Ded1 concentration compared to Ded1-hyperdependent mRNAs such as *CP-8.1* (analyzed above). As observed previously

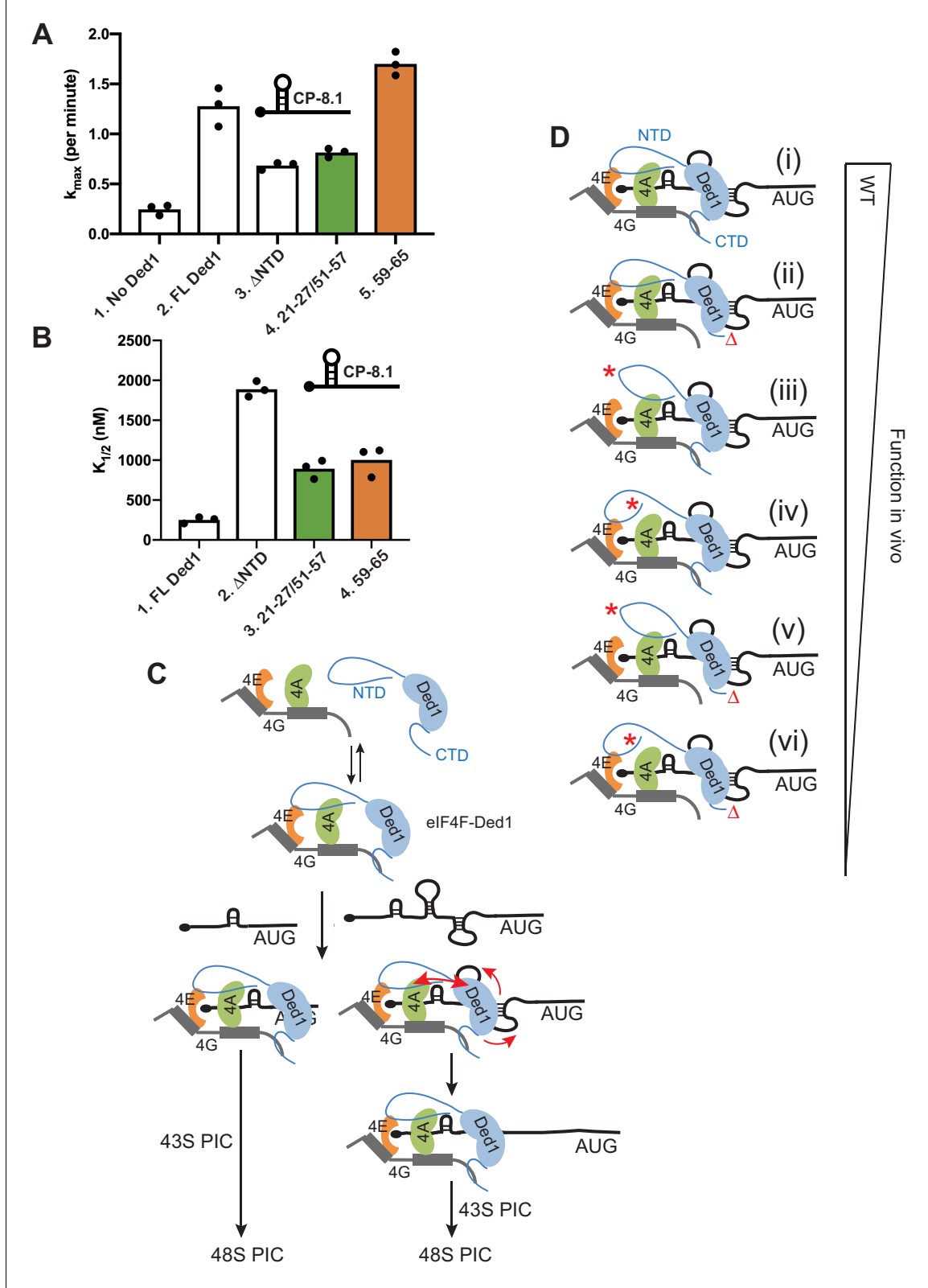

**Figure 9.** Disruption of Ded1 NTD interactions with eIF4A or eIF4E alter the kinetics of 48S assembly in the reconstituted system for a synthetic mRNA with Cap-proximal SL. (A–B) Kinetics of 48S PIC assembly was analyzed in reactions containing reconstituted 43S PICs, a radiolabeled capped reporter mRNA containing a cap-proximal SL of predicted ΔG of −8.1kcal/mol located five nt from the 5'end (depicted schematically), and different concentrations of mutant or WT Ded1 protein. Formation of 48S PICs was detected using a native gel mobility shift assay and measured as a function of

*Figure 9 continued on next page*

*Figure 9 continued*

time, allowing determination of observed rates at each Ded1 concentration. The maximum rates ($k_{max}$) (A) and Ded1 concentration at the half-maximal rate ($K_{1/2}$) (B) were determined from three replicate sets of assays and the individual values (black points) and mean values (bar heights) are plotted for experiments containing no Ded1, WT full-length Ded1 (FL Ded1), Ded1 lacking N-terminal residues 2–116 (ded1ΔNTD), or FL Ded1 harboring the indicated NTD substitutions that impair binding to eIF4A (green bars) or eIF4E (orange bars). (C) Model to account for the greater requirements for the Ded1-NTD interactions with eIF4A and eIF4E in stimulating translation of mRNAs with strong secondary structures versus relatively unstructured 5'UTRs. Ded1 interactions with each of the subunits of the eIF4F complex lowers the concentration of Ded1 required for its recruitment to the capped 5' ends of all mRNAs, where it can unwind structures that impede PIC attachment or scanning to the start codon. Structured mRNAs (*right*) require a more stable association between Ded1 and eIF4F for maximum Ded1 recruitment and, hence, are relatively more dependent on having both eIF4A and eIF4E contacts with the Ded1 NTD intact. Unwinding stable SL structures for the latter additionally requires the enhancement of Ded1 unwinding activity conferred by its interaction with eIF4A (red arrows on the *right*) to achieve maximum acceleration of PIC attachment or scanning. (D) Schematic summary of the relative importance of interactions of the Ded1 NTD with eIF4A and eIF4E and the Ded1 CTD with eIF4G, in stimulating Ded1 recruitment to mRNA and its RNA helicase activity. (i) In WT cells, Ded1's multiple interactions with the subunits of eIF4F enhance recruitment of Ded1 in complex with eIF4F to the m$^7$G cap to form a stable activated mRNP, which can subsequently recruit the 43S PIC and efficiently scan the 5'UTR to locate the AUG codon (not depicted). Eliminating the Ded1 CTD and its direct contact to eIF4G (ii, red Δ), confers a modest reduction in Ded1 function, whereas greater reductions are conferred by substitution mutations in the Ded1 NTD (red asterisks) that impair Ded1 interaction with eIF4E (iii) or eIF4A (iv). Even greater decreases in Ded1 function are seen on combining each of the Ded1 NTD substitutions with deletion of the Ded1 CTD (v–vi). The online version of this article includes the following source data and figure supplement(s) for figure 9:

Source data 1. CP-8.1 mRNA recruitment source data.
Figure supplement 1. Effects of Ded1 NTD substitutions that impair eIF4A or eIF4E binding on 48S PIC assembly on *RPL41A* mRNA in the yeast reconstituted system.
Figure supplement 1—source data 1. RPL41A mRNA recruitment source data.

(*Gupta et al., 2018*), FL Ded1 increased the $k_{max}$ for *RPL41A* mRNA by ~3 fold, and eliminating the Ded1 NTD by ΔNTD$_{1-116}$ did not impair this stimulation (*Figure 9—figure supplement 1A*, cols. 2–3); however, ΔNTD$_{1-116}$ increased the $K_{1/2}$ for Ded1 on *RPL41A* mRNA by nearly 70-fold (*Figure 9—figure supplement 1B*, cols. 2–3). As expected, neither the 21-27/51-57 double substitution nor the 59–65 substitution altered the increase in $k_{max}$ for *RPL41A* mRNA conferred by FL Ded1 (*Figure 9—figure supplement 1A*, cols. 4–5 vs. 2). While both substitutions increased the $K_{1/2}$ for Ded1 compared to FL-Ded1 (~7 fold for 21-27/51-57 and ~4 fold for the 59–65 substitution), these increases were much smaller than the ~70 fold increase produced by ΔNTD$_{1-116}$ (*Figure 9—figure supplement 1B*, cols. 3–4 vs. 2 cf. col.1; and *Figure 9—figure supplement 1D*, cols. 4–5 vs. 2 cf. col. 1). These last results suggest that strong interaction of either eIF4A or eIF4E with the Ded1-NTD is sufficient to greatly reduce the amount of Ded1 required for maximum acceleration of 48S PIC assembly on *RPL41A* mRNA. This contrasts with our findings on *CP-8.1* mRNA, where eliminating either of the interactions of the Ded1-NTD with eIF4A and eIF4E conferred an increase in the Ded1 $K_{1/2}$ only slightly smaller than that given by deleting the entire NTD. Thus, it appears that both interactions are needed simultaneously to enhance the function of the eIF4F-Ded1 complex on the more structured *CP-8.1* mRNA, whereas each interaction can contribute independently and additively on the less structured *RPL41A* mRNA.

## Discussion

In previous studies, segments of the N-terminus of Ded1 have been shown to be required for rapid cell growth at a reduced temperature (*Hilliker et al., 2011*; *Floor et al., 2016*; *Gao et al., 2016*), robust translation of a reporter mRNA in cell extracts (*Hilliker et al., 2011*), the stimulatory effect of eIF4A on Ded1 unwinding activity in vitro (*Gao et al., 2016*), and wild-type acceleration of 48S PIC assembly on particular mRNAs in the yeast reconstituted system (*Gupta et al., 2018*). Ded1 physically interacts with eIF4E (*Senissar et al., 2014*), but the eIF4E binding site within Ded1 was unknown. Importantly, it was also unclear whether Ded1's individual interactions with eIF4A or eIF4E are critical for Ded1 function in vivo in stimulating bulk protein synthesis and the translation of particular mRNAs with heightened Ded1-dependence conferred by 5'UTR structures. Finally, whereas interaction of the Ded1 CTD with eIF4G appears to enhance Ded1 unwinding function in vitro (*Putnam et al., 2015*; *Gao et al., 2016*), the ability of Ded1 to stimulate reporter translation in cell extracts (*Hilliker et al., 2011*) and 48S PIC assembly on particular mRNAs in a purified system (*Gupta et al., 2018*), it has been unclear whether this Ded1·eIF4G interaction is critical for Ded1

stimulation of translation in vivo. Our results fill in these important gaps in knowledge of how Ded1 stimulates translation initiation in living cells.

We have identified discrete, non-overlapping clusters of amino acids in the Ded1 NTD, as well as individual residues within these clusters, that appear to provide binding determinants for eIF4A or eIF4E, as their substitutions with alanine selectively reduce Ded1 binding to either GST-eIF4A or GST-eIF4E fusions in vitro. Binding determinants for eIF4A map between residues 21–27, including the highly conserved residue Arg-27, and also between residues 51–57, and appear to make independent, additive contributions to eIF4A binding. Binding determinants for eIF4E are located between residues 59–65, including Tyr-65, and between residues 83–89, including the highly conserved Trp-88, and appear to make concerted contributions to eIF4E binding. It is possible that additional binding determinants for eIF4E or eIF4A are located within the adjacent NTE extension of the helicase domain, which was not analyzed here, or within the Ded1 CTD (*Figure 1D*). Nevertheless, because the clustered NTD substitutions we identified selectively impair Ded1 binding to eIF4A or eIF4E in vivo at native levels of Ded1 expression, they provided us with the genetic tools needed to investigate whether Ded1's individual interactions with eIF4A or eIF4E are crucial for robust Ded1 function in vivo.

Armed with these Ded1 variants, we obtained evidence that disrupting Ded1-NTD interactions with either eIF4A or eIF4E reduces cell growth and bulk translation initiation, and preferentially impairs translation of reporters harboring 5′UTR SL structures in vivo. The Ded1 substitutions that impair interaction with eIF4A examined in combination with the *ded1-ts* mutation conferred a greater reduction in cell growth and an equal or somewhat greater impairment of the SL-containing reporters, but not a greater reduction in bulk polysomes, compared to the substitutions disrupting eIF4E binding to the Ded1 NTD. However, when introduced into otherwise WT Ded1, or the Ded1-Δ C variant, the substitutions perturbing eIF4A binding consistently impaired growth and both bulk and reporter translation more so than those impacting eIF4E binding. Thus, on balance, eIF4A interaction appears to be relatively more important than eIF4E interaction with the Ded1 NTD for strong translation initiation in vivo.

It was important to provide additional evidence that the Ded1 NTD substitutions impair translation initiation in vivo because they selectively impair Ded1 interactions with eIF4A or eIF4E rather than participating in some other aspect of Ded1 function. This was achieved for the substitutions that impair eIF4A binding by showing that overexpressing eIF4A mitigates the growth defects of NTD mutants *21–27* or *51–57*, which only partially impair eIF4A binding in vitro, but not of the double mutant *21-27/51-57* that appears to be incapable of binding eIF4A and, hence, should be refractory to a restoration of Ded1-eIF4A association by mass action. Nor did overexpressing eIF4A suppress the growth defects conferred by the *59-65/83-89* mutations that impair Ded1 binding to eIF4E. We used a different genetic approach to confirm that the latter NTD substitutions impair translation in vivo by diminishing Ded1-eIF4E association, by showing that overexpressing the 21-27/51-57 Ded1 variant, impaired for eIF4A association, behaves like overexpressed WT Ded1 and mitigates the phenotype of the eIF4E mutant *cdc33-1*, whereas overexpressing the 59-65/83-89 Ded1 variant selectively defective for eIF4E binding cannot rescue *cdc33-1* cells. While we cannot completely eliminate the possibility that the NTD mutations impair cell growth and translation in vivo by affecting some other aspect of Ded1 function with indirect consequences, these genetic suppression results provide confidence that Ded1-NTD interactions with eIF4A and eIF4E are physiologically important. It is noteworthy that the quadruple-cluster substitution mutation *21-27/51-57/59-65/83-89* that eliminates binding to both eIF4A and eIF4E confers a defect in cell growth greater than that seen for the two double-cluster substitutions that impair binding to only eIF4A or eIF4E, which is equivalent to that given by deleting the entire NTD (*Δ2–90*). These findings strongly suggest that eIF4A/eIF4E binding represents the crucial in vivo function of the Ded1 NTD.

The importance of the Ded1 CTD and its ability to interact with eIF4G in promoting translation initiation in vivo has been unclear, as its removal did not reduce growth in nutrient-replete cells; and while a deletion spanning the junction between the helicase core domain and the relatively conserved CTE was found to exacerbate the growth defects of two NTD mutations, a similar finding was not reported for a mutation affecting the more distal CTD domain (*Hilliker et al., 2011*). Here, we found that truncating the Ded1 CTD impairs cell growth, bulk polysome assembly, and translation of reporter mRNA in vivo only in Ded1 variants additionally impaired for their association with eIF4A or eIF4E via the Ded1 NTD. In view of recent findings that the last 14 residues of the CTD are crucial

for Ded1 binding to eIF4G in vitro (*Aryanpur et al., 2019*), our genetic findings imply that the interactions of Ded1 with each of the three subunits of eIF4F make independent, additive contributions to formation of the eIF4F·Ded1 complex and attendant stimulation of Ded1 helicase functions in promoting translation initiation. Interestingly, truncating the CTD to eliminate its ability to bind eIF4G confers resistance to cell growth inhibition by rapamycin, implicating the Ded1-CTD/eIF4G interaction in TORC1-mediated down-regulation of translation during nutrient starvation (*Aryanpur et al., 2019*) beyond its stimulatory function in nutrient-replete cells.

Using the reconstituted yeast system, we obtained evidence that individually disrupting Ded1-NTD interactions with either eIF4A or eIF4E impaired the ability of Ded1 to accelerate PIC assembly on a reporter mRNA harboring a cap-proximal SL. As we observed for a cap-proximal *FLUC* reporter in cells, Ded1 interaction with eIF4A was important for 48S PIC assembly on a cap-proximal SL reporter mRNA in vitro, as the Ded1 variant selectively impaired for eIF4A binding (21-27/51-57) reduced the maximum rate of PIC assembly ($k_{max}$) and also increased the amount of that variant required to achieve half-maximal rate stimulation ($K_{1/2}$) compared to WT Ded1. The Ded1 variant impaired for binding eIF4E (59-65) increased the Ded1 $K_{1/2}$ by a comparable amount, but did not reduce the $k_{max}$ for the cap-proximal SL reporter. While this might suggest a lesser requirement for Ded1 interaction with eIF4E versus eIF4A for rapid PIC assembly on this structured mRNA, the single-cluster variant 59–65 analyzed in vitro is not impaired for translation of the cap-proximal *FLUC* reporter in vivo, and we were unable to purify and analyze the double-cluster mutant 59-65/83-89 lacking both eIF4E binding determinants, which is compromised for translation of the cap-proximal promoter in vivo.

Interestingly, for the Ded1-hypodependent mRNA *RPL41A*, containing a 5' UTR of only 22nt lacking recognizable secondary structure, the Ded1 NTD$_{1-116}$, and hence interaction with both eIF4A and eIF4E, is dispensable for the modest ~2.5 fold increase in $k_{max}$ afforded by Ded1. Importantly, however, the $\Delta$NTD$_{1-116}$ variant (incapable of eIF4A/eIF4E interactions) requires an ~70 fold higher concentration compared to WT Ded1 to achieve the same half-maximal acceleration ($K_{1/2}$) on this mRNA. The 21-27/51-57 and 59–65 variants accelerate PIC assembly on *RPL41A* mRNA at ~10–20 times lower concentrations compared to $\Delta$NTD$_{1-116}$, indicating that interaction with either eIF4E or eIF4A by these substitution mutants is sufficient to increase the efficiency of Ded1 function on this mRNA compared to the absence of both interactions. For *CP-8.1* mRNA, by contrast, interaction with either eIF4E or eIF4A by the two aforementioned Ded1 variants confers only a small benefit, decreasing the $K_{1/2}$ by ~2 fold, whereas having both interactions intact in WT Ded1 lowers the $K_{1/2}$ by 7.6-fold, compared to $\Delta$N$_{1-116}$. Thus, while Ded1 interactions with eIF4A and eIF4E are important for Ded1 acceleration of PIC assembly on both mRNAs, the unstructured *RPL41A* mRNA appears to have a relatively smaller requirement for a full set of contacts between Ded1 and eIF4A/eIF4E in order for Ded1 to function at low concentrations. Moreover, *CP-8.1* additionally requires Ded1 contacts with eIF4A (impaired by substitutions 21-27/51-57) to overcome the SL structure and achieve maximum acceleration, even at high Ded1 concentrations.

One way to account for these differences between the *RPL41A* and *CP-8.1* mRNAs, depicted in the model shown in *Figure 9C*, would be to propose that the Ded1 interactions with the subunits of the eIF4F complex lowers the concentration of Ded1 required for its recruitment to the capped mRNA 5' end where it can unwind structures that impede PIC attachment or scanning to the start codon. This enhancement would apply to both unstructured and structured mRNAs, but structured mRNAs might require a more stable association between Ded1 and eIF4F for maximum Ded1 recruitment and, hence, be relatively more dependent on having both eIF4A and eIF4E contacts with the Ded1 NTD intact. Unwinding stable SL structures might additionally require the enhancement of Ded1 unwinding activity conferred by its interaction with eIF4A (*Gao et al., 2016*) to achieve maximum acceleration of PIC attachment or scanning (*Figure 9C*).

Together, our findings indicate that the Ded1 NTD interacts directly with eIF4E and eIF4A via adjacent, non-overlapping segments of the NTD, and that these interactions, in concert with the Ded1-CTD interaction with eIF4G, make independent, additive contributions to Ded1 function in stimulating translation initiation in vivo. As summarized schematically in *Figure 9D*, in all of our in vivo assays of cell growth and bulk or reporter mRNA translation, eliminating Ded1 interaction with eIF4G by deleting the Ded1 CTD had a less severe impact than did impairing eIF4A or eIF4E binding to the Ded1 NTD, and the eIF4A interaction with Ded1 appeared to be the most important overall. Envisioning all of these interactions occurring simultaneously on an mRNA suggests that they should

serve to tether Ded1 tightly to the activated eIF4F·mRNA complex and focus its helicase activity on the 5'UTR to unwind structures that impede PIC attachment or subsequent scanning (*Figure 9C*). Consistent with previous in vitro findings that interaction with eIF4A increases Ded1 unwinding activity (*Gao et al., 2016*), our results here in the reconstituted system suggest that the Ded1-NTD interaction with eIF4A is particularly important for enhancing Ded1 unwinding of the SL structure in CP-8.1 mRNA. It remains to be determined whether the Ded1 interaction with eIF4E serves primarily to stabilize formation of the eIF4F·Ded1 complex and localize it to mRNA 5' ends, or whether, it too stimulates Ded1 unwinding activity. In summary, we envision that Ded1-NTD interactions with both eIF4A and eIF4E, and Ded1-CTD binding to eIF4G, all help to assemble and stabilize the eIF4F-Ded1 complex at the 5' ends of mRNAs, which enhances initiation of nearly all mRNAs, whereas the Ded1-NTD/eIF4A interaction is additionally important to stimulate Ded1 unwinding of strong secondary structures in mRNA 5'UTRs (*Figure 9C*). Given the much greater abundance of eIF4A compared to eIF4E and eIF4G (*von der Haar and McCarthy, 2002*), the stimulation of Ded1 unwinding by eIF4A might also involve eIF4A molecules not tethered to eIF4F at the cap structure.

## Materials and methods

### Plasmids and yeast strains

All plasmids employed in this study are listed in *Table 1*. To create plasmids for bacterial expression of GST fusions to eIF4A1 and eIF4E, *TIF1* and *CDC33* coding sequences were amplified by PCR from the genomic DNA of WT yeast strain BY4741 (using primers listed in *Table 2*) and inserted between the BamHI and XhoI sites of pGEX4T1. pET-C-Ded1 cut with NdeI and XhoI was used to construct pSG49-pSG51 by inserting fragments encoding Ded1 residues 93–604 (pSG49), residues 1–561 (pSG50), or residues 93–561 (pSG51), PCR-amplified from pET-C-Ded1. Plasmids pSG52-pSG75 were derived from pET-C-Ded1 using the Agilent QuikChange Lightning Mutagenesis kit, according to the vendor's instructions, as were pSG76-pSG81 by mutagenesis of pET-N-Ded1. pSG1was constructed by inserting a fragment containing the *DED1* promoter (beginning 331 nt upstream of the AUG), entire coding sequence, coding sequence for the $myc_{13}$ epitope, and the *ADH1* terminator, PCR-amplified from the genomic DNA of yeast strain H3666, between into the SphI and SacI sites of *LEU2/CEN4/ARS1* vector YCplac111. pSG2, containing *ded1-ts-myc* was derived from pSG1 by mutagenesis with Agilent QuikChange Lightning Mutagenesis kit to introduce the T408I/W253R substitutions. pSG27-pSG43 were derived from pSG1; and pSG3-pSG26 were derived from pSG2, by QuikChange mutagenesis. pSG44-pSG46 were similarly derived from p4504/YEplac195-DED1.

All yeast strains employed in this study are listed in *Table 3*. Novel strains SGY1-SGY43 harboring plasmid pSG1-pSG43, were constructed by plasmid shuffling using 5-fluoroorotic acid (5-FOA) (*Boeke et al., 1987*) from strain yRP2799, as follows. Transformants of yRP2799 containing the relevant *LEU2* pSGY plasmid were selected on SC-Leu-Ura, patched on medium of the same composition, and replica-plated to SC-Leu containing 1 mg/ml 5-FOA. After 48–72 hr at 30°C, Ura⁻ segregants were purified from patches able to grow on 5-FOA medium by streaking for single colonies on SC-Leu.

### Yeast growth dilution-spot assays

Yeast strains were cultured in SC-Leu to $A_{600}$ of ~1.0, dilutions of $10^{-1}$, $10^{-2}$, $10^{-3}$, and $10^{-4}$ were prepared in sterile water, and 5 µL aliquots of undiluted and diluted yeast cells were spotted on SC-Leu medium and incubated at the temperatures indicated in the figure legends.

### Purification of recombinant GST fusion proteins expressed in *E. coli*

Agilent BL21-CodonPlus (DE3)-RIPL competent *E. coli* were transformed with pGEX4T1 or its derivatives containing the *TIF4631*, *TIF1* or *CDC33* coding sequences. Transformants were grown in 500 ml LB+Amp+Cam medium at 37°C to $A_{600}$ of ~0.5 and protein expression induced by adding IPTG to 1 mM and incubating at 18°C overnight with agitation. Harvested cells were washed with 50 ml phosphate-buffered saline (PBS) and lysed by sonication in the presence of 50 µg/mL lysozyme and 10% glycerol. Lysates were treated with 10% Triton X-100 at 4°C and cleared by centrifugation at 10,000xg. 500 µL of GE Healthcare Glutathione Sepharose 4B beads were added per 20 mL of lysate

**Table 1.** Plasmids and yeast alleles used in this study.

| Plasmid | Description | Source |
|---|---|---|
| YCplac111 | empty vector | *Gietz and Sugino, 1988* |
| pSG1 | *DED1-myc* in YCplac111 | This study |
| pSG2 | *ded1-ts-myc* in YCplac111 | This study |
| pSG3 | *ded1-ts/4–10-myc* in YCplac111 | This study |
| pSG4 | *ded1-ts/14–20-myc* in YCplac111 | This study |
| pSG5 | *ded1-ts/21–27-myc* in YCplac111 | This study |
| pSG6 | *ded1-ts/29–35-myc* in YCplac111 | This study |
| pSG7 | *ded1-ts/Δ40–47-myc* in YCplac111 | This study |
| pSG8 | *ded1-ts/51–57-myc* in YCplac111 | This study |
| pSG9 | *ded1-ts/59–65-myc* in YCplac111 | This study |
| pSG10 | *ded1-ts/68–74-myc* in YCplac111 | This study |
| pSG11 | *ded1-ts/83–89-myc* in YCplac111 | This study |
| pSG12 | *ded1-ts/90–95-myc* in YCplac111 | This study |
| pSG13 | *ded1-ts/21-27/29-35-myc* in YCplac111 | This study |
| pSG14 | *ded1-ts/21-27/51-57-myc* in YCplac111 | This study |
| pSG15 | *ded1-ts/59-65/83-89-myc* in YCplac111 | This study |
| pSG16 | *ded1-ts/21-27/51-57/59-65/83-89-myc* in YCplac111 | This study |
| pSG17 | *ded1-ts/Δ2–90-myc* in YCplac111 | This study |
| pSG18 | *ded1-ts,Δ562–604-myc* in YCplac111 | This study |
| pSG19 | *ded1-ts/21-27/51-57/59-65/83-89-/Δ562–604-myc* in YCplac111 | This study |
| pSG20 | *ded1-ts,Δ2–90,Δ562–604-myc* in YCplac111 | This study |
| pSG21 | *ded1-ts/Y21A-myc* in YCplac111 | This study |
| pSG22 | *ded1-ts/R27A-myc* in YCplac111 | This study |
| pSG23 | *ded1-ts/G28A-myc* in YCplac111 | This study |
| pSG24 | *ded1-ts/F56A/F57A-myc* in YCplac111 | This study |
| pSG25 | *ded1-ts/Y65A-myc* in YCplac111 | This study |
| pSG26 | *ded1-ts/W88A-myc* in YCplac111 | This study |
| pSG27 | *ded1-21-27-myc* in YCplac111 | This study |
| pSG28 | *ded1-51-57-myc* in YCplac111 | This study |
| pSG29 | *ded1-59-65-myc* in YCplac111 | This study |
| pSG30 | *ded1-83-89-myc* in YCplac111 | This study |
| pSG31 | *ded1-21-27/51-57-myc* in YCplac111 | This study |
| pSG32 | *ded1-59-65/83-89-myc* in YCplac111 | This study |
| pSG33 | *ded1-21-27/51-57/59-65/83-89-myc* in YCplac111 | This study |
| pSG34 | *ded1Δ2–90-myc* in YCplac111 | This study |
| pSG35 | *ded1Δ562–604-myc* in YCplac111 | This study |
| pSG36 | *ded1-21-27/Δ562–604-myc* in YCplac111 | This study |
| pSG37 | *ded1-51-57/Δ562–604-myc* in YCplac111 | This study |
| pSG38 | *ded1-59-65/Δ562–604-myc* in YCplac111 | This study |
| pSG39 | *ded1-83-89/Δ562–604-myc* in YCplac111 | This study |
| pSG40 | *ded1-21-27/51-57/Δ562–604-myc* in YCplac111 | This study |
| pSG41 | *ded1-59-65/83-89/Δ562–604-myc* in YCplac111 | This study |
| pSG42 | *ded1-21-27/51-57/59-65/83-89/Δ562–604-myc* in YCplac111 | This study |
| pSG43 | *ded1Δ2–90,Δ562–604-myc* in YCplac111 | This study |

*Table 1 continued on next page*

*Table 1 continued*

| Plasmid | Description | Source |
|---|---|---|
| p1992/YEplac195 | empty vector | *Gietz and Sugino, 1988* |
| p3333/pBAS3432 | *TIF1* in YEplac195 | *Neff and Sachs, 1999* |
| p3351 | *CDC33* in YEplac195 | *de la Cruz et al., 1997* |
| p4504/YEplac195-DED1 | *DED1* in YEplac195 | *de la Cruz et al., 1997* |
| pSG44 | *ded1-21-27/51-57* in YEplac195 | This study |
| pSG45 | *ded1-59-65/83-89* in YEplac195 | This study |
| pSG46 | *ded1Δ2–90* in YEplac195 | This study |
| p6053/pFJZ342 | *RPL41A* 5'UTR with 23 CAA repeats inserted (91nt long) in YCplac33 | *Sen et al., 2015* |
| p6058/pFJZ669 | *RPL41A* 5'UTR with cap-proximal SL with ΔG of −8.1 kcal/mol in YCplac33 | *Sen et al., 2015* |
| p6062/pFJZ623 | *RPL41A* 5'UTR with cap-distal SL with ΔG of −3.7 kcal/mol in YCplac33 | *Sen et al., 2015* |
| p2917/pGEX-4T1 | empty vector | GE Healthcare 28954549 |
| pGEX-4G1 | *TIF4631* in pGEX-4T1 | *Mitchell et al., 2010* |
| pSG47 | *TIF1* in pGEX-4T1 | This study |
| pSG48 | *CDC33* in pGEX-4T1 | This study |
| pET22b | empty vector | EMD Millipore 69744–3 |
| p5946/pET-C-Ded1 | *DED1* in pET22b | *Hilliker et al., 2011* |
| pSG49 | *ded1(93-604)-His* in pET22b | This study |
| pSG50 | *ded1(1-561)-His* in pET22b | This study |
| pSG51 | *ded1(93-561)-His* in pET22b | This study |
| pSG52 | *ded1-4-10-His* in pET22b | This study |
| pSG53 | *ded1-14-20-His* in pET22b | This study |
| pSG54 | *ded1-21-27-His* in pET22b | This study |
| pSG55 | *ded1-29-35-His* in pET22b | This study |
| pSG56 | *ded1-Δ40–47-His* in pET22b | This study |
| pSG57 | *ded1-51-57-His* in pET22b | This study |
| pSG58 | *ded1-59-65-His* in pET22b | This study |
| pSG59 | *ded1-68-74-His* in pET22b | This study |
| pSG60 | *ded1-83-89-His* in pET22b | This study |
| pSG61 | *ded1-90-95-His* in pET22b | This study |
| pSG62 | *ded1-21-27/29-35-His* in pET22b | This study |
| pSG63 | *ded1-21-27/51-57-His* in pET22b | This study |
| pSG64 | *ded1-21-27/68-74-His* in pET22b | This study |
| pSG65 | *ded1-51-57/59-65-His* in pET22b | This study |
| pSG66 | *ded1-59-65/83-89-His* in pET22b | This study |
| pSG67 | *ded1-Y21A-His* in pET22b | This study |
| pSG68 | *ded1-R27A-His* in pET22b | This study |
| pSG69 | *ded1-G28A-His* in pET22b | This study |
| pSG70 | *ded1-R51A-His* in pET22b | This study |
| pSG71 | *ded1-F56A/F57A-His* in pET22b | This study |
| pSG72 | *ded1-R62A-His* in pET22b | This study |
| pSG73 | *ded1-Y65A-His* in pET22b | This study |
| pSG74 | *ded1-R83A-His* in pET22b | This study |
| pSG75 | *ded1-W88A-His* in pET22b | This study |
| pET-N-Ded1 | *His-DED1* in pET15b | *Gupta et al., 2018* |

*Table 1 continued*

| Plasmid | Description | Source |
|---|---|---|
| pET-SUMO-ded1ΔN | *His-ded1Δ2–117* in pET-SUMO | *Gupta et al., 2018* |
| pSG76 | *His-ded1-21* in pET15b | This study |
| pSG77 | *His-ded1-51* in pET15b | This study |
| pSG78 | *His-ded1-59* in pET15b | This study |
| pSG79 | *His-ded1-83* in pET15b | This study |
| pSG80 | *His-ded1-21,51* in pET15b | This study |
| pSG81 | *His-ded1-59,83* in pET15b | This study |
| pGIBS-LYS | *lysA* | ATCC #87482 |
| pGIBS-TRP | *trpCDEF* | ATCC #87485 |
| pGIBS-PHE | *pheB* | ATCC #87483 |

and incubated with rotation overnight at 4°C. The beads were washed twice with 5 ml PBS in 10% glycerol and treated with 5000 gel units of NEB Micrococcal Nuclease in the presence of 2 mM calcium chloride in 1 ml PBS. The reaction was stopped by addition of EDTA to 5 mM, followed by washing the beads with 1 ml of 500 mM sodium chloride in PBS/glycerol. The beads were washed two more times with 1 ml PBS in 10% glycerol and the washed were stored at −80°C. SDS-polyacrylamide gel electrophoresis (SDS-PAGE), followed by Coomassie Blue staining, was employed to estimate the amounts of GST-tagged proteins bound to the glutathione-agarose beads by comparison to known amounts of bovine serum albumin.

## In vitro translation of Ded1 polypeptides

Promega TNT Quick Coupled Transcription/Translation System was employed for in vitro translation of Ded1 proteins for GST pull-down assays. Perkin Elmer EasyTag L-[$^{35}$S]-Methionine was used for radioactive labeling following the vendor's instructions.

## In vitro GST pull down assays

GST pull down reactions were conducted as follows. Aliqouts of 10 µL of GST or GST-tagged bait proteins bound to glutathione-agarose beads were combined with 8 µL of pull-down buffer (40 mM Tris, pH 8.0, 0.01% IGEPAL CA630, 2 mM DTT, 50 mM NaCl, 20% glycerol, and 5 mM MgCl$_2$), 0.5 µL of 5 mg/ml bovine serum albumin, 0.5 µL of ThermoFisher Scientific RNase A/T1 mix and 1 µL of [$^{35}$S]-labeled Ded1 protein. The reactions were incubated at 4°C with end-over-end rotation overnight. The beads were washed three times with 30 µL of pull-down buffer without glycerol and proteins eluted with 20 µL 15 mM Glutathione [pH 8.0]. The input proteins and eluates were resolved by SDS-PAGE and visualized by fluorography, as follows. Gels were fixed in 50% methanol and 15% acetic acid, stained with Coomassie Blue, treated with Amersham Amplify Fluorographic Reagent, dried under vacuum and exposed to X-ray film for 18–72 hr at −80°C. After developing, the films and gels were scanned, and band intensities analyzed by NIH ImageJ software. Pull down efficiency was calculated as intensity of pulled down band divided by the corresponding input band intensity and converted to percentages. For statistical analyses, three or four replicates were performed. Unpaired student's *t* test was employed to determine p-values. GraphPad software was used to graph the data as dot plots.

## Coimmunoprecipitation and western blots using yeast WCEs

For coimmunoprecipitation analysis, transformants of H3446 containing the relevant plasmid-borne *DED1-myc* alleles or empty vector were cultured in 100 mL of SC-Leu-Trp to A$_{600}$ of ~0.8, harvested by centrifugation, washed with coimmunoprecipitation (CoIP) buffer (100 mM Tris-HCl pH 7.5, 100 mM sodium chloride, 0.1% NP-40, 1 mM DTT, 1 mM PMSF, 1 × Complete Protease Inhibitor Mix tablets without EDTA [Roche]) and lysed by vortexing with glass beads. Lysates were cleared by centrifugation and total protein amount determined colorimetrically using BioRad Bradford Reagent. The WCEs extracts were diluted to 5 mg/mL in CoIP buffer and aliquots of 1 mg (200 µL) were

**Table 2.** Primer sequences used in this study.

| Primer | 5'-Sequence-3' | Source |
|---|---|---|
| SG1/Ded1-13xmyc-F | TTTTTTGCATGCCCAAAGGTGTTCTTATGTAGTGACACCGAT | This study |
| SG2/Ded1-13xmyc-R | TTTTTTGAGCTCTTACCCTGTTATCCCTAGCGGATCTGCCGG | This study |
| SG134/W253R-F | AATTTACTTATAGATCCAGGGTCAAGGCCTGCGTC | This study |
| SG135/W253R-R | GACGCAGGCCTTGACCCTGGATCTATAAGTAAATT | This study |
| SG136/T408I-F | CAATTGATCTGCCATTCTCTTAATTTCGACAAAGATCAAAGTCAAA | This study |
| SG137/T408I-R | TTTGACTTTGATCTTTGTCGAAATTAAGAGAATGGCAGATCAATTG | This study |
| SG3/Cdc33-F | TTTTTTGGATCCTCCGTTGAAGAAGTTAGCAAG | This study |
| SG4/Cdc33-R | TTTTTTCTCGAGTTACAAGGTGATTGATGGTTG | This study |
| SG132/Tif1-F | TTTTTTGGATCCATGTCTGAAGGTATTACTGA | This study |
| SG133/Tif1-R | TTTTTTCTCGAGTTAGTTCAACAAAGTAGCGA | This study |
| SG5/pETded1(93-561)-F | GGGGGTCATATGCATGTCCCAGCTCCAAGAAA | This study |
| SG6/pETded1(93-561)-R | TTTTTTCTCGAGGCCTCCGGCCTTACGGTAAT | This study |
| SG7/pETded1(93-604)-F | TTTTTTCATATGCATGTCCCAGCTCCAAGAAAC | This study |
| SG8/pETded1(93-604)-R | TTTTTTCTCGAGCCACCAAGAAGAGTTGTTTGA | This study |
| SG9/pETded1(1-561)-F | TGTGTGCATATGGCTGAACTGAGCGAACAAGTG | This study |
| SG10/pETded1(1-561)-R | TTTTTTCTCGAGGCCTCCGGCCTTACGGTAAT | This study |
| SG41/pET4-s | GTGAGGAGGAACATAACCATTCTCGTTGTTGTCGTT GATGCTTAAAGCTGCCGCTGCTGCGGCCGCTTCAGC CATATGTATATCTCCTTCTTAAAGTTAAACAAAATTATTT | This study |
| SG42/pET4-as | AAATAATTTTGTTTAACTTTAAGAAGGAGATATACAT ATGGCTGAAGCGGCCGCAGCAGCGGCAGCTTTAAG CATCAACGACAACAACGAGAATGGTTATGTTCCTCCTCAC | This study |
| SG43/y4-s | GGAGGAACATAACCATTCTCGTTGTTGTCGTTGATG CTTAAAGCTGCCGCTGCTGCGGCCGCTTCAGCCAT AATATGAAATGCTTTTCTTGTTGTTCTTACGGA | This study |
| SG44/y4-as | TCCGTAAGAACAACAAGAAAAGCATTTCATATTAT GGCTGAAGCGGCCGCAGCAGCGGCAGCTTTAAG CATCAACGACAACAACGAGAATGGTTATGTTCCTCC | This study |
| SG45/14 s | CTTGGTTTTCCTCTTAAGTGAGGAGGAACATAA GCAGCCGCGGCGGCGGCGGCGATGCTTAAATT TTGCACTTGTTCGCTCAGTTCAGCCA | This study |
| SG46/14-as | TGGCTGAACTGAGCGAACAAGTGCAAAATTTAA GCATCGCCGCCGCCGCCGCGGCTGCTTATGTT CCTCCTCACTTAAGAGGAAAACCAAG | This study |
| SG47/21 s | TTGCTACTGTTATTTCTGGCACTTCTTGGTTTT CCTGCTGCGGCAGCAGCAGCAGCACCATTCTC GTTGTTGTCGTTGATGCTTAAATTTTGCACTTG | This study |
| SG48/21-as | CAAGTGCAAAATTTAAGCATCAACGACAACAACG AGAATGGTGCTGCTGCTGCTGCCGCAGCAGGAA AACCAAGAAGTGCCAGAAATAACAGTAGCAA | This study |
| SG49/29 s | GTTGTAGCCGCCGTTGTTGTTATTGTAGTTGCTA CTGTTAGCTGCTGCAGCTGCTGCTGCTCCTCTTA AGTGAGGAGGAACATAACCATTCTCGTTGTTG | This study |
| SG50/29-as | CAACAACGAGAATGGTTATGTTCCTCCTCACTTAA GAGGAGCAGCAGCAGCTGCAGCAGCTAACAGTAG CAACTACAATAACAACAACGGCGGCTACAAC | This study |
| SG51/51 s | ACCACCACGACGGTTGTTGCTAGCGGCGGCGG CAGCGGCAGCGCCACCGTTGTAGCCGCCGTTG | This study |
| SG52/51-as | CAACGGCGGCTACAACGGTGGCGCTGCCGCTG CCGCCGCCGCTAGCAACAACCGTCGTGGTGGT | This study |
| SG53/68 s | CGTTAGATCTGCTGCCACCGTTGGCTGCAGCGG CGGCAGCAGCGTTGCCGTAACCACCACGACGG | This study |

*Table 2 continued on next page*

*Table 2 continued*

| Primer | 5′-Sequence-3′ | Source |
|---|---|---|
| SG54/68-as | CCGTCGTGGTGGTTACGGCAACGCTGCTGCCGCC GCTGCAGCCAACGGTGGCAGCAGATCTAACG | This study |
| SG55/83 s | TTGGAGCTGGGACATGTTTGCCATCGGCCGCTGC AGCAGCAGCAGCGCCGTTAGATCTGCTGCCACCGTTGT | This study |
| SG56/83-as | ACAACGGTGGCAGCAGATCTAACGGCGCTGCTGCTG CTGCAGCGGCCGATGGCAAACATGTCCCAGCTCCAA | This study |
| SG57/90 s | GATCTCGGCCTTTTCGTTTCTTGCAGCTGCGGCAGC TGCGGCAGCGATCCATCTACCACCAGAACGG | This study |
| SG58/90-as | CCGTTCTGGTGGTAGATGGATCGCTGCCGCAGCTG CCGCAGCTGCAAGAAACGAAAAGGCCGAGATC | This study |
| SG105/40del-s | AAATAACAGTAGCAACAACGGTGGCCGTGGCG | This study |
| SG106/40del-as | CGCCACGGCCACCGTTGTTGCTACTGTTATTT | This study |
| SG303/59 s | TTCCACCGAAGAAACCACCGTTGCCGGCAGCAG CAGCAGCGGCGGCGCTAAAGAAGCTGCCACCGCCACGGC | This study |
| SG304/59-as | GCCGTGGCGGTGGCAGCTTCTTTAGCGCCGCCGC TGCTGCTGCTGCCGGCAACGGTGGTTTCTTCGGTGGAA | This study |
| SG305/59on51-s | CACCGAAGAAACCACCGTTGCCGGCAGCAGCAGC AGCGGCGGCGCTAGCGGCGGCGGCAGCGGCAG | This study |
| SG306/59on51-as | CTGCCGCTGCCGCCGCCGCTAGCGCCGCCGCTG CTGCTGCTGCCGGCAACGGTGGTTTCTTCGGTG | This study |
| SG195/R27A-s | CTTCTTGGTTTTCCTGCTAAGTGAGGAGGAACATAACCATTCTCG | This study |
| SG196/R27A-as | CGAGAATGGTTATGTTCCTCCTCACTTAGCAGGAAAACCAAGAAG | This study |
| SG197/R51A-s | CTAAAGAAGCTGCCAGCGGCAGCGCCACCGTTGTAGCC | This study |
| SG198/R51A-as | GGCTACAACGGTGGCGCTGCCGCTGGCAGCTTCTTTAG | This study |
| SG199/R62A-s | GAAACCACCGTTGCCGTAAGCAGCAGCACGGTTGTTGCTAAAGAAG | This study |
| SG200/R62A-as | CTTCTTTAGCAACAACCGTGCTGCTGCTTACGGCAACGGTGGTTTC | This study |
| SG201/R83A-s | CCATCGATCCATCTACCAGCAGCAGCGCCGTTAGATCTGCTGCC | This study |
| SG202/R83A-as | GGCAGCAGATCTAACGGCGCTGCTGCTGGTAGATGGATCGATGG | This study |
| SG161/ydelC-s | CCGTAAGGCCGGAGGCGGTGAACAAAGCTAAT | This study |
| SG162/ydelC-as | ATTAGCTTTTGTTCACCGCCTCCGGCCTTACGG | This study |
| SG265/y2-90del-s | CTGGGACATGTTTGCCATCCATAATATGAAATGCTTTTCTTGTTGTTC | This study |
| SG266/y2-90del-as | GAACAACAAGAAAAGCATTTCATATTATGGATGGCAAACATGTCCCAG | This study |
| SG233/Y21A-s | AAGTGAGGAGGAACAGCACCATTCTCGTTGTTGTCGTTGATGC | This study |
| SG234/Y21A-as | GCATCAACGACAACAACGAGAATGGTGCTGTTCCTCCTCACTT | This study |
| SG235/F56F57-s | CACGACGGTTGTTGCTAGCGGCGCTGCCACCGCCACGGC | This study |
| SG236/F56F57-as | GCCGTGGCGGTGGCAGCGCCGCTAGCAACAACCGTCGTG | This study |
| SG241/Y65A-s | CCACCGTTGCCGGCACCACCACGACGGTTGT | This study |
| SG242/Y65A-as | ACAACCGTCGTGGTGGTGCCGGCAACGGTGG | This study |
| SG243/W88A-s | CATGTTTGCCATCGATCGCTCTACCACCAGAACGGC | This study |
| SG244/W88A-as | GCCGTTCTGGTGGTAGAGCGATCGATGGCAAACATG | This study |
| SG249/G28A-s | TGGCACTTCTTGGTTTTGCTCTTAAGTGAGGAGGA | This study |
| SG250/G28A-as | TCCTCCTCACTTAAGAGCAAAACCAAGAAGTGCCA | This study |
| FZ158/Fluc-F | ATG GAA GAC GCC AAA AAC ATA AAG | *Sen et al., 2015* |
| FZ159/Fluc-R | TTA CAA TTT GGA CTT TCC GCC CTT | *Sen et al., 2015* |
| act1-F | TGTGTAAAGCCGGTTTTGCC | *Zeidan et al., 2018* |
| act1-R | GATACCTCTCTTGGATTGAGCTTC | *Zeidan et al., 2018* |

**Table 3.** Yeast strains used in this study.

| Strain | Description | Source |
|---|---|---|
| F2041/ yRP2799 | *MATa his3Δ1 leu2Δ0 lys2Δ0 met15Δ0 ura3Δ0 ded1Δ::kanMX4 pRP1560 (DED1 URA3)* | *Hilliker et al., 2011* |
| H3666 | *MATa his3Δ1 leu2Δ0 met15Δ0 ura3Δ0 DED1-myc$_{13}$ HIS3* | *Dong et al., 2005* |
| F729/ BY4741 | *MATa his3Δ1 leu2Δ0 met15Δ0 ura3Δ0* | Research Genetics |
| H4436[*] | *MATa ade2 his3 leu2 trp1 ura3 pep4::HIS3 tif4631::leu2hisG tif4632::ura3 pEP88 (TIF4631-HA-Bam TRP1 CEN4)* | *Park et al., 2011* |
| F694 | *MATa cdc33::LEU2 ade2 his3 leu2 trp1 ura3 p(CDC33, TRP1, ARS/CEN4)* | *Altmann and Trachsel, 1989* |
| F695 | *MATa cdc33::LEU2 ade2 his3 leu2 trp1 ura3 p(cdc33-4-2, TRP1, ARS/CEN4)* | *Altmann and Trachsel, 1989* |
| F696 | *MATa cdc33::LEU2 ade2 his3 leu2 trp1 ura3 p(cdc33-1, TRP1, ARS/CEN4)* | *Altmann and Trachsel, 1989* |
| SGY1 | *MATa his3Δ1 leu2Δ0 lys2Δ0 met15Δ0 ura3Δ0 ded1Δ::kanMX4 pSG1 (DED1-myc LEU2)* | This study |
| SGY2 | *MATa his3Δ1 leu2Δ0 lys2Δ0 met15Δ0 ura3Δ0 ded1Δ::kanMX4 pSG2 (ded1-ts-myc LEU2)* | This study |
| SGY3 | *MATa his3Δ1 leu2Δ0 lys2Δ0 met15Δ0 ura3Δ0 ded1Δ::kanMX4 pSG3 (ded1-ts/4–10-myc LEU2)* | This study |
| SGY4 | *MATa his3Δ1 leu2Δ0 lys2Δ0 met15Δ0 ura3Δ0 ded1Δ::kanMX4 pSG4 (ded1-ts/14–20-myc LEU2)* | This study |
| SGY5 | *MATa his3Δ1 leu2Δ0 lys2Δ0 met15Δ0 ura3Δ0 ded1Δ::kanMX4 pSG5 (ded1-ts/21–27-myc LEU2)* | This study |
| SGY6 | *MATa his3Δ1 leu2Δ0 lys2Δ0 met15Δ0 ura3Δ0 ded1Δ::kanMX4 pSG6 (ded1-ts/29–35-myc LEU2)* | This study |
| SGY7 | *MATa his3Δ1 leu2Δ0 lys2Δ0 met15Δ0 ura3Δ0 ded1Δ::kanMX4 pSG7 (ded1-ts/Δ40–47-myc LEU2)* | This study |
| SGY8 | *MATa his3Δ1 leu2Δ0 lys2Δ0 met15Δ0 ura3Δ0 ded1Δ::kanMX4 pSG8 (ded1-ts/51–57-myc LEU2)* | This study |
| SGY9 | *MATa his3Δ1 leu2Δ0 lys2Δ0 met15Δ0 ura3Δ0 ded1Δ::kanMX4 pSG9 (ded1-ts/59–65-myc LEU2)* | This study |
| SGY10 | *MATa his3Δ1 leu2Δ0 lys2Δ0 met15Δ0 ura3Δ0 ded1Δ::kanMX4 pSG10 (ded1-ts/68–74-myc LEU2)* | This study |
| SGY11 | *MATa his3Δ1 leu2Δ0 lys2Δ0 met15Δ0 ura3Δ0 ded1Δ::kanMX4 pSG11 (ded1-ts/83–89-myc LEU2)* | This study |
| SGY12 | *MATa his3Δ1 leu2Δ0 lys2Δ0 met15Δ0 ura3Δ0 ded1Δ::kanMX4 pSG12 (ded1-ts/90–95-myc LEU2)* | This study |
| SGY13 | *MATa his3Δ1 leu2Δ0 lys2Δ0 met15Δ0 ura3Δ0 ded1Δ::kanMX4 pSG13 (ded1-ts/21-27/29-35-myc LEU2)* | This study |
| SGY14 | *MATa his3Δ1 leu2Δ0 lys2Δ0 met15Δ0 ura3Δ0 ded1Δ::kanMX4 pSG14 (ded1-ts/21-27/51-57-myc LEU2)* | This study |
| SGY15 | *MATa his3Δ1 leu2Δ0 lys2Δ0 met15Δ0 ura3Δ0 ded1Δ::kanMX4 pSG15 (ded1-ts/59-65/83-89-myc LEU2)* | This study |
| SGY16 | *MATa his3Δ1 leu2Δ0 lys2Δ0 met15Δ0 ura3Δ0 ded1Δ::kanMX4 pSG16 (ded1-ts/21-27/51-57/59-65/83-89-myc LEU2)* | This study |
| SGY17 | *MATa his3Δ1 leu2Δ0 lys2Δ0 met15Δ0 ura3Δ0 ded1Δ::kanMX4 pSG17 (ded1-ts,Δ2–90-myc LEU2)* | This study |
| SGY18 | *MATa his3Δ1 leu2Δ0 lys2Δ0 met15Δ0 ura3Δ0 ded1Δ::kanMX4 pSG18 (ded1-ts,Δ562–604-myc LEU2)* | This study |
| SGY19 | *MATa his3Δ1 leu2Δ0 lys2Δ0 met15Δ0 ura3Δ0 ded1Δ::kanMX4 pSG19 (ded1-ts/21-27/51-57/59-65/83-89-myc/ Δ562–604-myc LEU2)* | This study |
| SGY20 | *MATa his3Δ1 leu2Δ0 lys2Δ0 met15Δ0 ura3Δ0 ded1Δ::kanMX4 pSG20 (ded1-ts,Δ2–90,Δ562–604-myc LEU2)* | This study |
| SGY21 | *MATa his3Δ1 leu2Δ0 lys2Δ0 met15Δ0 ura3Δ0 ded1Δ::kanMX4 pSG21 (ded1-ts/Y21A-myc LEU2)* | This study |
| SGY22 | *MATa his3Δ1 leu2Δ0 lys2Δ0 met15Δ0 ura3Δ0 ded1Δ::kanMX4 pSG22 (ded1-ts/R27A-myc LEU2)* | This study |
| SGY23 | *MATa his3Δ1 leu2Δ0 lys2Δ0 met15Δ0 ura3Δ0 ded1Δ::kanMX4 pSG23 (ded1-ts/G28A-myc LEU2* | This study |
| SGY24 | *MATa his3Δ1 leu2Δ0 lys2Δ0 met15Δ0 ura3Δ0 ded1Δ::kanMX4 pSG24 (ded1-ts/F56A/F57A-myc LEU2)* | This study |
| SGY25 | *MATa his3Δ1 leu2Δ0 lys2Δ0 met15Δ0 ura3Δ0 ded1Δ::kanMX4 pSG25 (ded1-ts/Y65A-myc LEU2)* | This study |
| SGY26 | *MATa his3Δ1 leu2Δ0 lys2Δ0 met15Δ0 ura3Δ0 ded1Δ::kanMX4 pSG26 (ded1-ts/W88A-myc LEU2)* | This study |
| SGY27 | *MATa his3Δ1 leu2Δ0 lys2Δ0 met15Δ0 ura3Δ0 ded1Δ::kanMX4 pSG27 (ded1-21-27-myc LEU2)* | This study |
| SGY28 | *MATa his3Δ1 leu2Δ0 lys2Δ0 met15Δ0 ura3Δ0 ded1Δ::kanMX4 pSG28 (ded1-51-57-myc LEU2)* | This study |
| SGY29 | *MATa his3Δ1 leu2Δ0 lys2Δ0 met15Δ0 ura3Δ0 ded1Δ::kanMX4 pSG29 (ded1-59-65-myc LEU2)* | This study |
| SGY30 | *MATa his3Δ1 leu2Δ0 lys2Δ0 met15Δ0 ura3Δ0 ded1Δ::kanMX4 pSG30 (ded1-83-89-myc LEU2)* | This study |
| SGY31 | *MATa his3Δ1 leu2Δ0 lys2Δ0 met15Δ0 ura3Δ0 ded1Δ::kanMX4 pSG31 (ded1-21-27,51–57-myc LEU2)* | This study |
| SGY32 | *MATa his3Δ1 leu2Δ0 lys2Δ0 met15Δ0 ura3Δ0 ded1Δ::kanMX4 pSG32 (ded1-59-65,83–89-myc LEU2)* | This study |

*Table 3 continued on next page*

Table 3 continued

| Strain | Description | Source |
|--------|-------------|--------|
| SGY33 | MATa his3Δ1 leu2Δ0 lys2Δ0 met15Δ0 ura3Δ0 ded1Δ::kanMX4 pSG33 (ded1-21-27/51-57/59-65/83-89-myc-myc LEU2) | This study |
| SGY34 | MATa his3Δ1 leu2Δ0 lys2Δ0 met15Δ0 ura3Δ0 ded1Δ::kanMX4 pSG34 (ded1Δ2–90-myc LEU2) | This study |
| SGY35 | MATa his3Δ1 leu2Δ0 lys2Δ0 met15Δ0 ura3Δ0 ded1Δ::kanMX4 pSG35 (ded1Δ562–604-myc LEU2) | This study |
| SGY36 | MATa his3Δ1 leu2Δ0 lys2Δ0 met15Δ0 ura3Δ0 ded1Δ::kanMX4 pSG36 (ded1-21-27/Δ562–604-myc LEU2) | This study |
| SGY37 | MATa his3Δ1 leu2Δ0 lys2Δ0 met15Δ0 ura3Δ0 ded1Δ::kanMX4 pSG37 (ded1-51-57/Δ562–604-myc LEU2) | This study |
| SGY38 | MATa his3Δ1 leu2Δ0 lys2Δ0 met15Δ0 ura3Δ0 ded1Δ::kanMX4 pSG38 (ded1-59-65/Δ562–604-myc LEU2) | This study |
| SGY39 | MATa his3Δ1 leu2Δ0 lys2Δ0 met15Δ0 ura3Δ0 ded1Δ::kanMX4 pSG39 (ded1-83-89/Δ562–604-myc LEU2) | This study |
| SGY40 | MATa his3Δ1 leu2Δ0 lys2Δ0 met15Δ0 ura3Δ0 ded1Δ::kanMX4 pSG40 (ded1-21-27/51-57/Δ562–604-myc LEU2) | This study |
| SGY41 | MATa his3Δ1 leu2Δ0 lys2Δ0 met15Δ0 ura3Δ0 ded1Δ::kanMX4 pSG41 (ded1-59-65/83-89/Δ562–604-myc LEU2) | This study |
| SGY42 | MATa his3Δ1 leu2Δ0 lys2Δ0 met15Δ0 ura3Δ0 ded1Δ::kanMX4 pSG42 (ded1-21-27/51-57/59-65/83-89-myc/Δ562–604-myc LEU2) | This study |
| SGY43 | MATa his3Δ1 leu2Δ0 lys2Δ0 met15Δ0 ura3Δ0 ded1Δ::kanMX4 pSG43 (ded1-Δ2–90,Δ562–604-myc LEU2) | This study |

*The plasmid in this strain contains *TIF4631-HA* under the control of the *TIF4632* promoter and transcription terminator, as described previously (*Tarun and Sachs, 1996*) modified to insert a BamHI site at the start codon (*Park et al., 2011*).

immunoprecipitated with 5 µg Roche anti-myc antibody (clone 9E10). After 2 hr of rotation at 4°C, 50 µL ThermoFisher Scientific Dynabeads Protein G were added and rotation continued for 2 hr more. Using a magnetic rack, the beads were washed three times with 150 µL CoIP buffer. The beads were resuspended in 100 µL 2x Laemmli buffer and boiled for 5 min. SDS-PAGE and Western blot were employed to detect the immunoprecipitated proteins, using the following antibodies: mouse monoclonal anti-HA antibody (12C5; Roche Cat# 11 666 606 001), mouse monoclonal anti-c-Myc antibody (9E10; Roche Cat# 11 667 203 001), rabbit polyclonal anti-eIF4A/Tif1 (kindly provided by Patrick Linder), rabbit polyclonal anti-eIF4E/Cdc33 (kindly provided by J.E.G. McCarthy), and rabbit polyclonal anti-Ded1 antibody (kindly provided by Dr. Tien-Hsien Chang). The immune complexes were visualized by enhanced chemiluminescence.

For Western blot analysis of yeast WCEs, trichloroacetic acid (TCA) precipitation was employed to extract total protein (*Reid and Schatz, 1982*), extracts were resolved by. SDS-PAGE, and Western blots were probed using the antibodies listed above, as well as rabbit polyclonal anti-Hcr1 antibodies (*Valásek et al., 2001*).

## Luciferase reporter assays

Two-ml cultures of yeast transformants harboring the relevant reporter plasmids were harvested by centrifugation and the cell pellets were washed with 500 µL PBS and lysed by vortexing with glass beads in 200 µL PBS supplemented with 1 × Complete Protease Inhibitor Mix tablets without EDTA [Roche]. Lysates were cleared by centrifugation, diluted 1:100 in PBS supplemented with 1 × Complete Protease Inhibitor Mix tablets without EDTA and luciferase was assayed using Promega Luciferase Assay System and a luminometer. Total protein concentration was measured colorimetrically using Bradford Reagent (BioRad) and known amounts of bovine serum albumin as standards. Luciferase units were normalized by the total protein amounts.

## Polysome fractionation and analysis of mRNAs

For analyzing polysomes profiles, 200 mL of cells were cultured in the appropriate medium at 30°C and shifted to 36°C for 3 hr or to 15°C for 1 hr, and harvested at $A_{600} \approx 1$. Cells were treated with 50 µg/ml cycloheximide for 5 min prior to harvesting. WCEs were prepared by resuspending the cell pellet with an equal volume of breaking buffer (20 mM Tris-HCl, pH 7.5, 50 mM KCl, 10 mM MgCl₂, 1 mM dithiothreitol [DTT], 5 mM NaF, 1 mM phenylmethylsulfonyl fluoride [PMSF], 1 × Complete Protease Inhibitor Mix tablets without EDTA [Roche], 1 U/µl SUPERase-In RNase inhibitor) and vortexing with glass beads, followed by two cycles of centrifugation for 10 min at 15,000 rpm at 4°C. 20 $A_{260}$ units of cleared lysate were resolved on 10–50% (w/w) sucrose gradients by centrifugation at

39,000 rpm for 160 min at 4°C in a Beckman SW41Ti rotor. Gradients were fractionated with a Brandel Fractionation System with continuous monitoring at 254 nm with an ISCO UV detector. Fractions (0.7 mL) were precipitated overnight at −20°C by adding 1.5 volumes of RNA precipitation mix (95% ethanol, 5% sodium acetate pH 5.2), and centrifuged at 15,000 rpm for 30 min. 5 ng of 'spike-in RNA' was added to each fraction to control for differences in RNA recovery: a mixture of in vitro transcribed capped *Bacillus subtilis* mRNAs (*Arava et al., 2003*): 80 pg/µL lysA (prepared from plasmid pGIBS-LYS; ATCC#87482), 160 pg/µl trpCDEF (prepared from plasmid pGIBS-TRP; ATCC#87485), and 320 pg/µl of pheB (prepared from plasmid pGIBS-PHE; ATCC #87483). This mixture of in vitro transcribed mRNAs was a kind gift from Dr. Neelam Sen (*Sen et al., 2019*). Pellets were washed in 1 mL cold 80% ethanol, dried in a speed vacuum, resuspended in 100 µL TE buffer (10 mM Tris-HCl pH 8.0, 1 mM EDTA) and treated with 10 Kunitz units of DNase I. Qiagen RNEasy Mini kit was used to extract RNA and resuspended in 25 µL RNase-free water.

For RT-qPCR analysis of mRNA abundance in each gradient fraction, reverse transcription was carried out using SuperScript III First-Strand cDNA Synthesis SuperMix (Invitrogen) and 2 µL RNA purified from each gradient fractions. qPCR was carried out using Brilliant III Ultra-Fast SYBR Green Master Mix (Agilent) in a Mx3000P System (Stratagene) and the oligonucleotide pairs listed in *Table 2*. Normalization and analyses were performed as described in *Chiu et al., 2010* and *Sen et al., 2015*.

## mRNA recruitment assays in the yeast reconstituted system

### Preparation and purification of mRNAs, charged initiator tRNA and translation initiation factors

Plasmids for in vitro run-off mRNA and initiator tRNA transcription are described in *Acker et al., 2007*; *Mitchell et al., 2010*; *Gupta et al., 2018*. *RPL41A* mRNA represents the full-length native transcript. CP-8.1 mRNA has an unstructured 5'UTR comprised of CAA repeats with a stem-loop inserted five nt from the cap appended to the body of *RPL41A*. The mRNAs were transcribed, purified and capped as described in *Gupta et al., 2018*. Initiator tRNA was transcribed, methionylated and purified as described previously (*Yourik et al., 2017*). mRNAs were capped ($m^7$GpppG) in vitro using vaccinia virus D1/D12 capping enzyme with either $\alpha$-$^{32}$P radiolabeled GTP (Perkin Elmer) or unlabeled GTP (for pulse-chase) (*Mitchell et al., 2010*). Eukaryotic initiation factors, 40S ribosomal subunits, and WT Ded1 and its mutant derivatives were expressed and purified as described previously (*Acker et al., 2007*; *Mitchell et al., 2010*; *Munoz et al., 2017*; *Gupta et al., 2018*).

The Ded1 proteins bearing N-terminal His$_6$-tags (Genscript) were expressed from the appropriate pET22b vectors in *E. coli* strain BL21(DE3) RIL CodonPlus cells (Agilent) by culturing cells in LB medium at 37°C to A$_{600}$ of ~0.5, cooled to 20°C, and induced by addition of IPTG to 0.5 mM for 16 hr. Cells were harvested and resuspended in 35 mL lysis buffer (10 mM HEPES-KOH, pH-7.4, 200 mM KCl, 0.1% IGEPAL CA-630, 10 mM imidazole, 10% glycerol, 10 mM 2-mercaptoethanol, and cOmplete protease inhibitor cocktail (Roche)) and lysed using a French Press. The cell lysate was treated with DNaseI (1 U/mL) for 20 min on ice and the KCl concentration was adjusted to 500 mM. His$_6$-Ded1 proteins were affinity-purified using a nickel column (5 ml His-Trap column, GE Healthcare). Fractions containing the Ded1 proteins were collected, the glycerol concentration was adjusted to 30%, and the Ded1 was further purified by phosphocellulose chromatography (P11, Whatman). Purified Ded1 proteins were dialyzed against dialysis buffer (10 mM HEPES-KOH, pH 7.4, 200 mM KOAc, 50% Glycerol, 2 mM DTT) and stored at −80°C.

### mRNA recruitment kinetics

mRNA recruitment kinetics were performed using a native gel shift assay as described previously (*Mitchell et al., 2010*; *Yourik et al., 2017*). This assay determines the apparent rates (k$_{app}$) and maximal rates (k$_{max}$) of recruitment as well as the concentration of a factor required to achieve half-maximal rate of recruitment (K$_{1/2}$). Briefly, PICs were assembled with 300 nM eIF2, 0.5 mM GDPNP·Mg$^{2+}$, 200 nM Met-tRNA$_i^{Met}$, 1 µM eIF1, 1 µM eIF1A, 300 nM eIF5, 300 nM eIF4B, 300 nM eIF3, 75 nM eIF4E·eIF4G, 7 µM eIF4A (15 µM for *SFT2* mRNA) and 30 nM 40S subunits in 1X Recon buffer (30 mM HEPES-KOH, pH 7.4, 100 mM KOAc, 3 mM Mg(OAc)$_2$, and 2 mM DTT). The concentration of Ded1 was varied in the reactions from 0 to 1000 nM for WT Ded1 and 0–6000 nM for the Ded1 variants. Complexes were assembled at 26°C for 10 min and reactions were initiated by addition of 15

nM $^{32}$P-m$^7$G capped mRNA and 5 mM ATP·Mg$^{2+}$. Reactions were quenched by addition of 600–750 non-radiolabeled m$^7$G-mRNA at appropriate times (*Mitchell et al., 2010*; *Yourik et al., 2017*) and 48S PICs were separated from the free mRNA on a 4% non-denaturing PAGE gel. The fraction of the mRNA recruited to the 48S PIC at each time-point was calculated using ImageQuant software (GE Healthcare). Data were fitted with a single exponential rate equation and hyperbolic equations (KaleidaGraph software (Synergy)) to determine the apparent rates ($k_{app}$) and maximal rates ($k_{max}$) of recruitment as well as the concentration of a factor required to achieve half-maximal rate of recruitment ($K_{1/2}$). Bar-graph data representations were prepared using Prism 7 (GraphPad).

## Multiple sequence alignments

T-Coffee (www.ebi.ac.uk/Tools/msa/tcoffee) was used to align Ded1 amino acid sequences from fungal and animal species. Saccharomyces Genome Database (https://www.yeastgenome.org/) was the source of Ded1 sequences from the genus Saccharomyces. Other sequences were retrieved from NCBI Protein (https://www.ncbi.nlm.nih.gov/protein) and UniProt (https://www.uniprot.org/). The results were reformatted with MVic (www.ebi.ac.uk/Tools/msa/mview) and WebLogo (weblogo.berkeley.edu).

## Acknowledgements

We thank Roy Parker, Angie Hilliker, Alan Sachs, Patrick Linder, and Michael Altmann for gifts of yeast strains or plasmids, Patrick Linder, JEG McCarthy, and Tien-Hsien Chang for gifts of antibodies, and Neelam Sen for spike-in RNAs. We thank Neelam Sen, Fan Zhang, Swati Gaikwad, other members of our laboratories, Nick Guydosh, and Tom Dever for helpful comments and suggestions. This work was supported in part by the Intramural Research Program of the National Institutes of Health.

# Additional information

## Competing interests

Alan G Hinnebusch: Reviewing editor, *eLife*. The other authors declare that no competing interests exist.

## Funding

| Funder | Grant reference number | Author |
|---|---|---|
| National Institutes of Health | Intramural Research Program | Suna Gulay<br>Neha Gupta<br>Jon R Lorsch<br>Alan G Hinnebusch |

The funders had no role in study design, data collection and interpretation, or the decision to submit the work for publication.

## Author contributions

Suna Gulay, Conceptualization, Data curation, Formal analysis, Investigation, Methodology, Writing - original draft; Neha Gupta, Data curation, Formal analysis, Investigation, Writing - review and editing; Jon R Lorsch, Conceptualization, Formal analysis, Supervision, Project administration, Writing - review and editing; Alan G Hinnebusch, Conceptualization, Formal analysis, Supervision, Funding acquisition, Writing - original draft, Project administration, Writing - review and editing

## Author ORCIDs

Suna Gulay (iD) https://orcid.org/0000-0002-4672-144X
Jon R Lorsch (iD) http://orcid.org/0000-0002-4521-4999
Alan G Hinnebusch (iD) https://orcid.org/0000-0002-1627-8395

**Decision letter and Author response**
Decision letter https://doi.org/10.7554/eLife.58243.sa1
Author response https://doi.org/10.7554/eLife.58243.sa2

## Additional files

### Supplementary files

• Transparent reporting form

### Data availability

All data generated or analysed during this study are included in the manuscript and supporting files. Source data files have been provided for Figures 9 and 9-supplement 1.

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
