## [Decision Letter]

[Editors' note: this paper was reviewed by Review Commons.]

**Acceptance summary:**

The authors present convincing evidence that the interactions of eIF4A and eIF4E with Ded1 are critical in vivo in yeast. They delineated the locations in Ded1 of the binding sites for eIF4A and eIF4E. They provided strong evidence for the function of Ded1 in promoting translation in vivo and in vitro in a purified system. This study raises the interesting question as to why the mammalian Ded1 homolog, DDX3X is involved in many functions other than translation in the cell, such as miRNA biogenesis, mRNA nucleo-cytoplasmic transport, pre-mRNA splicing, and others, while in yeast it functions strictly in translation.

---

## [Author Response]

Reviewer #1 (Evidence, reproducibility and clarity):The manuscript by Gulay et al. describes a study into the role of contacts between the RNA helicase Ded1 and components of the translation initiation factor 4F complex during eukaryotic protein synthesis. Using the purified yeast factors and targeted mutagenesis, the authors delineate binding sites for eIF4E and eIF4A in the Ded1 N-terminal domain. They then study the importance of these binding sites for growth and translation initiation in yeast in vivo, and in a reconstituted in vitro system, showing that both sites enhance the functioning of Ded1 during translation initiation, with subtly different effects.In my view the study is overall expertly done and relies on a broad portfolio of techniques that explores the importance of the two binding sites from a variety of angles. I think the discussion and the conclusions drawn are quite conservative and certainly do not overinterpret the data, however I see it as a downside that the discussion so strictly adheres to a static view of a fully formed eIF4F complex as the substrate for Ded1 binding, as exemplified by the scheme in Figure 9C. On a number of counts, the authors' data suggest a much more flexible process, and a discussion that reflects this could enhance the manuscript. The first four points below make specific suggestions around which this could be based.1) Figure 5 suggests that mutations in the eIF4E binding site reduce interaction with eIF4E without reducing the interaction with eIF4G. Since the eIF4E:eIF4G interaction is very stable, this makes it unlikely that Ded1 interacts with a fully formed eIF4F complex (otherwise the pull-down of eIF4G should also be affected). Moreover, Ded1 was never detected in the initial cap-pull downs which led to the characterisation of eIF4F, again arguing that its interaction may be restricted to particular times or states of the initiation machinery.

This is a good point and we have revised the text to include the possible interpretation that Ded1 can interact separately with the individual subunits of eIF4F in addition to the intact complex. However, this would be odds with the expectation that the majority of Ded1 is associated with eIF4F in cells, based on estimates of the binding constant for Ded1 association with eIF4F and the cellular concentrations of these factors (Gao et al., 2016). Rather, it seems plausible that weakening the association of Ded1 with eIF4E would lead to a specific reduction in eIF4E coimmunoprecipitation with *Ded1-myc* owing to dissociation of eIF4E from eIF4G during the extensive washing steps involved in the experiment, notwithstanding the stable interaction of eIF4E with eIF4G. The absence of Ded1 in the initial cap pull-down experiments that recovered eIF4F can be explained by the known high off-rate of Ded1 from eIF4F (Gao et al., 2016).

2) The authors' data contain some evidence for regulatory interactions within Ded1. First, deletion of the CTD appears to enhance eIF4A binding but reduce eIF4E binding (Figure 1C). I note that the magnitude of the increase in eIF4A binding is similar to the decrease in eIF4E binding, yet the authors describe the first as "comparable" but the second as "reduced" – it would be useful to see statistics here.

We had indeed neglected to add the statistics to Figures 1C-D, which we rectified by revising these panels of Figure 1. While the reduction in Ded1 binding to GST-eIF4E is significantly reduced for the Ded1-ΔC variant (P=0.03); the apparent increase in Ded1-ΔC binding to GST-eIF4A is not significant (P=0.10), which underlies our conclusion that *ΔC* impairs binding to eIF4E but not eIF4A.

Second, I note that the 14-20 mutant appears to suppress the ded1-ts slow growth phenotype (Figure 3B, lane 3 at 34C) and also shows a slight increase in eIF4E binding (Figure 2C).

We find that this apparent suppression was not evident in several independent transformants and added this information to the text.

The apparent increase in binding of Ded1 variant 14-20 to GST-eIF4E is not significant (P=0.08).

3) Figure 8C suggests distinct effects of Ded1 binding to eIF4A (important for translating transcripts with both cap-proximal and cap-distal stem loops) and eIF4E (only important for transcripts with cap-distal stem loops). This is not consistent with the model in Figure 9C, where the different interactions only contribute additive binding contacts with eIF4F.

Figure 8C shows that impairing Ded1 binding to either eIF4E (with Ded1-59-65/83-89 substitutions) or eIF4A (with 21-27/51-57 substitutions) significantly impairs translation of both cap-proximal and cap-distal SL reporters, just to a lesser extent for the cap-proximal reporter on weakening eIF4E-Ded1 interaction with Ded1-59-65/83-89. This difference in magnitude of the translation defects is consistent with our model in Figure 9C if we stipulate that impairing the Ded1-eIF4E interaction by the 59-65/83-89 substitutions has a smaller impact on assembly of the eIF4F-Ded1 complex compared to that conferred by the Ded1 21-27/51-57 substitutions and that robust translation of the cap-proximal SL reporter is more dependent on eIF4F-Ded1 association compared to the cap-distal reporter. Text to this effect has been added to the discussion of the model in Figure 9.

4) The distinction between effects on attachment of Ded1 to eIF4F vs helicase activity is only superficially discussed. Figure 9A and B show that both eIF4E and eIF4A are important for attachment of Ded1 to eIF4F (as both binding mutants affect the K_1/2_), but only eIF4A is important for stem loop unwinding (as only the eIF4A binding site affects the K_max_). This is discussed by the authors. I think the fact that on an unstructured stem loop both binding site mutants still affect the K_1/2_ but not the K_max_ further supports this – presumably this means that the Ded1/eIF4F interaction is always more efficient when supported by the additional eIF contacts, whereas Ded1 helicase activity is sufficient on its own on unfolded constructs but requires support from eIF4A on stem-loop containing transcripts.

We thank the reviewer for this insight, and we have added new Figure 9C and explanatory text to represent this interpretation.

5) A last point where clarification would be useful is on the polysome profiles in Figure 8. Translation initiation defects often lead to the appearance of a shoulder on the monosome peak, which is usually interpreted as mRNAs with one translating ribosome and initiating 40S subunit (e.g.PMID 12471154 uses this extensively). None of the profiles shown in this study show such a shoulder, can the authors explain why this is?

The absence of halfmers can be explained if reducing Ded1 function by impairing its association with eIF4F primarily reduces attachment of PICs to the mRNA or impedes the migration of scanning PICs to the AUG codon for assembly of stable 48S PICs (with Met-tRNA_i_ base paired to AUG) that can withstand the hydrostatic pressures of sedimentation through sucrose. Indeed, our recent 40S profiling of *ded1* mutants indicates the occurrence of both such defects in PIC attachment and scanning.

Moreover, although the polysome parts of the profiles clearly decay for the new ded1 mutants, the 80S peak does not seem to increase proportionally especially in the 59-65/83-89 mutant. Based on the "mRNP" peak which should be a useful internal standard for the numbers of cells lysed, this mutant seems to have lost most of its ribosome content.

We appreciate the point, but don’t think that the proportion of total A_260_ present in the free mRNP peak can be assumed to be constant between WT and *ded1* mutants in which the proportion of mRNA in mRNPs might well increase as the result of reduced polysome assembly.

Lastly, I'm surprised that the ded1-ts mutant on its own seems to have virtually no effect on translation even at the restrictive temperature of 36C which the legend to Figure 8 indicates was used here (cf. Figure 8A and Figure 8—figure supplement 2).

It cannot be said from the data in Figure 8A that the *ded1-ts* mutation has no impact on polysome assembly because the WT *DED1* profile isn’t shown here. However, we had analyzed isogenic *DED1^+^* cells in parallel and found, as expected, a significantly greater P/M ratio (3.90 +/- 0.11) compared to that shown in the figure for *ded1-ts* (2.39 +/- 0.10). This information has been added to the Figure 8A legend.

Reviewer #1 (Significance):The work adds significant conceptual detail to our understanding of the role of Ded1 in translation initiation. On the path to understanding the mechanism of translation in any detail, this is essential work. Thanks to the thorough and methodologically broad approach used, the work strongly enhances our understanding of a central biological process, and since the interaction sites are highly conserved throughout eukaryotes, the work done here in yeast informs our understanding of general eukaryotic biology. Like any good study that forays into new territory, the work raises as many questions as it answers – one immediate one being questions around the order of events during complex formation in vivo as well as regulation of events during complex assembly. With alterations to the discussion based on my points above, I think this side would be presented more clearly. There is a strong history of fundamental studies into translation factors influencing and informing important fields including cancer (e.g. PMID 28653885) and other diseases (30038383, 30290184) and understanding fundamental translation factor biology is also becoming important for manipulating translation in bioprocessing and synthetic biology (27760840, 30615942). Thus, I have no doubt that the present study will be of interest to a wide readership.

As noted above, we altered the discussion based on several of the reviewer’s points.

Reviewer #2 (Evidence, reproducibility and clarity):In this manuscript entitled "Distinct interactions of eIF4A and eIF4E with RNA helicase Ded1 stimulate translation in vivo", Gulay et al. identify separate amino acid clusters within the NTD of Ded1 required for interaction with eIF4E and eIF4A. They demonstrate that disruption of these clusters impairs cell growth, steady-state polysome levels and preferentially affects translation of an mRNA reporter with 5' UTR structure versus one with no/less structure. Disrupting Ded1 NTD interactions with eIF4E or eIF4E affected kinetics of 48S assembly on a synthetic mRNA with cap-proximal structure.– If possible, can the authors provide us with a sense of the type of affinities that Ded1 has for eIF4E and eIF4A?

Based on K_1/2_ values determined by Gao et al. from kinetic analysis of unwinding of a model substrate, the affinity of Ded1 for eIF4A was estimated to be in the range of 45-800 nM, depending on whether Ded1 or eIF4A are present alone (800 nM) or eIF4A is complexed with RNA (45 nM) (Gao et al., 2016). Unfortunately, we have no estimate for the affinity of Ded1 for eIF4E.

– Figure 3D – there appears to be two bands for Ded1. Can the authors comment on these?

Unfortunately, we cannot comment definitively on this point.

It looks like in lane 8, there is reduced expression of D2-90 compared to the mutants in lanes 6 and 7. Perhaps quantification from several gels can be provided under the panel.

We added to Figure 3D the requested quantification of the Ded1/Hcr1 ratios determined from multiple replicates, and indicated in the legend to Figure 3D that the particular replicate shown here was atypical in suggesting a reduced level of the *ded1-ts-Δ2-90* product vs. the parental *ded1-ts* product.

– Figure 6. How would the 4E binding mutants behave in this assay?

We had examined this previously and found no effect of eIF4A overexpression on the growth defect of the *ded1-ts/59-65/83-89* allele, which is specifically defective for binding eIF4E. We have added this control observation in a new Supplementary figure to Figure 6 (Figure 6—figure supplement 1) and mentioned it in the text, which supports our interpretation of results in Figure 6B that eIF4A suppresses the growth defect of *ded1* alleles that are partially defective for binding eIF4A by restoring Ded1-eIF4A association through mass action.

– The sentence "Hence it is conceivable that W88 and R27 mediate conserved interactions between Ded1 and its Ddx3 homologs in animals with eIF4E and eIF4A, respectively." is speculative. Please remove or rephrase to focus only on what is in the paper – the yeast data.

We have removed the sentence.

– Figure 8. Why was the Cap-proximal SL reporter used in D, rather than the cap-distal SL which showed a greater effect in panel C? Do the authors have data for the cap-distal SL.

In panel C, both reporters showed a marked reduction in expression for the two mutants impaired for binding eIF4A or eIF4E analyzed in panel D. Unfortunately, we don’t have data for the cap-distal SL reporter for the experiment in panel D, which is very labor-intensive.

– Figure 9. The nomenclature used for the mutants is different than what is used in the rest of the figures. Also, the numbers don't align properly with the columns in Panels A and B.

These small errors in the figure have been fixed.

– Figure 9 A, B. Can the authors provide the raw data for the results leading to the data in Panels A and B? Perhaps as a supplemental Figure.

This source data has been provided as a supplementary excel file.

Reviewer #3 (Evidence, reproducibility and clarity):Summary:This manuscript by Gulay, et al. focuses on the function of Ded1, a DEAD-box RNA helicase, in translation initiation, specifically its interaction with the eIF4F translation complex. The authors identify small mutations in Ded1 that affect its interactions with eIF4A and eIF4E, and they utilize these mutants to examine the importance of the interactions for cell growth and translation in budding yeast using previously established in vivo and in vitro techniques. For the most part, the data and conclusions of the manuscript are solid; however, there are a few points at which further control experiments would be helpful and/or opportunities were missed to increase confidence in the results. There are also a few issues with interpretation/discussion that should be addressed.Major comments:1) In Figure 2, have the authors considered whether there might be competition for 4A and 4E binding to Ded1? This would not be consistent with their overall model, but it would explain the effects on 4A binding in the 51-57 mutant vs. 51-57/59-65 mutant. A competition binding experiment could reasonably be done with recombinant proteins in hand.

Such competition is not possible in this experiment as only GST-eIF4A or GST-eIF4E are present in the binding reactions.

2) In Figure 2—figure supplement 4, the single point mutants should be tested for effects on binding of both 4A and 4E as controls. This is especially relevant for the mutants in the 51-65 region that appear to have cross-antagonistic effects (point 1).

We have determined that R27A, which impairs binding to GST-eIF4A, does not affect binding to GST-eIF4E, as expected from the fact that the 21-27 substitution

(encompassing R27A) specifically impairs interaction with GST-eIF4A (Figure 2B-C). We had also determined that R51A does not impair binding to GST-eIF4E and that R83A does not affect binding to GST-eIF4A, indicating that neither of these Ded1 substitutions affect either interaction. These additional results have been added to Figure 2—figure supplement 4. While we have not tested the Y65A and W88A substitutions for their effects on binding to GST-eIF4A, it seems highly unlikely that they would impair this interaction considering that the 59-65 and 83-89 clustered Ala substitutions, which encompass Y65A and W88A, respectively, did not did so (Figure 2B).

3) Along these same lines, the Y21A mutant should be tested for 4D binding and possibly other defects (4G, enzymatic defects) since it has no 4A binding defect but significant in vivo phenotypes.

As already shown in Figure 2C, the 21-27 clustered Ala substitution, which encompasses Y21A, did not affect binding to GST-eIF4E, making it highly unlikely that Y21A would do so. We feel that understanding the molecular basis of the growth defect conferred by the Y65A substitution is outside the scope of this report.

4) The rationalization for the focus from Figure 3 on in using ded1-ts as a sensitized background is not very well-described. The mutants by themselves seem to have phenotypes (e.g. Figure 3C), why not use these without the possibly confounding effects of another mutation?

When we began the study, we weren’t sure if mutations specifically impairing either eIF4E or eIF4A would have a marked phenotype on their own, noting that deletion of the Ded1 CTD that eliminates the eIF4G binding site has no effect on cell growth in otherwise WT cells. Hence, we chose the standard genetic practice of using a sensitive background and evaluating synthetic phenotypes in combination with *ded1-ts*.

Along these same lines, the polysome profile of the ded1-ts control strain in Figure 8A is surprisingly normal (higher P/M than the DED1 strain in Figure 8—figure supplement 3) given that this strain was previously shown to have a substantial defect under less restrictive conditions (Sen et al., 2015).

As noted above for reviewer 2, we had analyzed isogenic *DED1^+^* cells in parallel and found, as expected, a significantly greater P/M ratio (3.90 +/- 0.11) compared to that shown in the figure for *ded1-ts* (2.39 +/- 0.10). This information has been added to the Figure 8A legend.

5) The suppression experiments in Figures 6 and 7 are good genetic evidence. However, the authors' interpretation of the 21-27/51-57 double mutant does not seem to fit the data, since the interpretation relies on a complete ablation of 4A binding, while this mutant still appears to interact at a reduced level in vitro and in vivo.

We disagree that it can be concluded that the Ded1 21-27/51-57 double mutant retains residual binding to eIF4A, as the difference in binding to GST-eIF4A between this variant and complete deletion of the Ded1 NTD (ΔN) in Figure 2B is not statistically significant. In addition, the residual CoIP of eIF4A with the myc-tagged Ded1 harboring 21-27/51-57 could easily be mediated by eIF4G, whose coimmunoprecipitation is unaffected by these NTD substitutions.

Further, an additional control needed in Figure 6 is to express hcTIF1 in the 4E binding mutant (59-65/83-89) – if the authors are correct, it should not rescue growth. Also, have the authors attempted hcDED1 in a TIF1/TIF2 mutant? That would complement the other suppression experiments nicely. Finally, do the suppressions also rescue polysome and/or scanning defects as in Figure 8?

As mentioned above for reviewer 2, the predicted result was obtained as hc*TIF1* does not suppress the growth defect of the 59-65/83-89 double mutation, and has been included in a new supplementary figure (Figure 6—figure supplement 1). We were unsure of the reviewer’s objective in asking us to test for suppression of a *TIF1/TIF2* mutant by hc*DED1,* and unclear about what mutant should be examined. A new supplementary figure (Figure 8—figure supplement 5) and explanatory text has been added providing evidence that hc*TIF1* mitigates the reduction in *LUC* reporter expression conferred by the *ts/21* mutation (that allows residual eIF4A/Ded1 interaction) but not by the *ts/21,51* mutation (that abolishes eIF4A interaction), nor by the *ts/59,83* mutation that impairs eIF4E interaction (included as a negative control). Rescue by hc*CDC33* was not attempted because eIF4E overexpression confers a slow-growth phenotype, which is why we reverted to overexpression of Ded1 in the *cdc33-1* mutant in the genetic analysis of Figure 7.

6) The manuscript relies heavily on genetic arguments, but some caution is warranted, given that these arguments are often somewhat indirect. This is especially true given that no attempt is described at making mutations in 4A and 4E that do not interact with Ded1. I do not think constructing such mutants is required, but they should temper their conclusions and offer caveats throughout, especially when discussing the model in Figure 9C.

A statement has been added to the Discussion that, while it is impossible to rule out additional effects of the *ded1* mutations beyond impairing Ded1 interaction with eIF4A or eIF4E, the genetic suppression experiments in Figures 6-7 provide strong evidence in favor of this interpretation.

7) In regard to the model, the authors claim that the 4A interaction may be "more important" than the 4E one, largely based on the in vitro translation assays in Figure 9. However, only the single (59-65) 4E-binding mutant was used in these assays compared to the double (21-27/51-57) 4A-binding mutant. Either the 51-57/83-89 double mutant should be tested, or the authors need to modify their claim about the relative importance of binding to the two factors.

We agree and have modified our claim accordingly in the Discussion.

Minor comments:1) The focus on how the 4A/4E mutants "exacerbate" defects caused by deletion of the CTD is a fairly weak point. The effects seem largely additive rather than necessarily illustrating a major aspect of the interaction with eIF4F. I would recommend moving Figure 4, for instance, to the supplement and somewhat reducing the prominence of commentary about the CTD in the manuscript. Additionally, it is not completely accurate that no studies have shown an in vivo effect of the Ded1-CTD and/or the Ded1-eIF4G interaction – Hilliker et al. showed some in vivo effects of the CTD deletion, Senissar et al. showed genetic interactions on growth with ded1 mutants and eIF4G, and Aryanpur et al., 2019, recently showed a significant in vivo effect of the CTD, linked to eIF4G interaction, in certain conditions.

We prefer to leave Figure 4 in its current position because we consider it significant that the deletion of the Ded1 CTD confers a growth defect only in the presence of the Ded1 NTD mutations that impair eIF4E or eIF4A binding, ie. we observe synthetic growth phenotypes between these NTD and *ΔC* mutations. This provides important evidence for additive effects of separate Ded1 interactions with eIF4A, eIF4E and eIF4G in assembly of an eIF4F·Ded1 complex that enhances Ded1 stimulation of translation. However, we have revised the Discussion to better explain the significance of this synthetic genetic interaction.

Re-examining the literature cited by the reviewer did not reveal any published findings that contradict our conclusion that the in vivo importance of the Ded1 CTD in stimulating translation at native levels of Ded1 expression has been unclear. Hilliker et al., 2011, observed a slight exacerbation of the Cs^-^ phenotype of a double NTD mutation in combination with a deletion that spans the boundary between the core helicase and CTE of Ded1; and they found that a deletion in the CTD impairs Ded1 stimulation of translation in vitro, but it had no effect on cell growth even at low temperature where defects in Ded1 function are most readily observed. Senissar et al., 2014, examined no mutations in the Ded1 CTD. Aryanpur et al., 2019, recently uncovered an interesting role for the Ded1-CTD in TORC1-mediated down-regulation of translation initiation (rather than stimulation) during nutrient starvation. We have revised the Discussion to more thoroughly discuss our findings on the CTD in relation to these published findings.

2) The strain background used in the pull-downs in Figure 5 is described differently in the text compared to the figure legend and Table 2. First, tagged TIF4631 appears to be expressed from a plasmid, not from the eIF4G2 promoter. Second, was wild-type Ded1 present in these cells? If so, its presence in the pull-downs could be confounding in interpreting the results.

The strain descriptions are consistent throughout, but we have added explanatory information in the Results and Table indicating that the strain contains deletions of *TIF4631* and *TIF4632* and contains *TIF4631-HA* on a plasmid under the control of the *TIF4632* promoter and transcription terminator, as described previously by Tarun and Sachs. We have also added text indicating that the presence of WT *DED1* in the strain could be intensifying the effects of the mutations in the myc-tagged *ded1* alleles on association with eIF4F components by competition for binding the latter.

3) In a number of the Western blots, the protein levels in the exposures shown are fairly saturated and do not appear to be in a linear range (e.g. Figure 6C). It is difficult to be sure the expression levels are actually similar if this is the case.

As noted above for reviewer 2, we have added quantification of Ded1 protein levels from multiple replicates to Figure 3D. For Figure 4B, there is no question that the Ded1 variants containing the point mutations are expressed as well or better than the parental WT or *ΔC* proteins, establishing the key objective of the Western analysis. For Figure 6C, the main purpose of the analysis was to confirm that eIF4A is comparably overexpressed in the key *ts/21-27/51-57* mutant (lane 12) versus the other mutants (lanes 2, 4, 6, 8, 10), which we feel was achieved, noting that the eIF4A signals are invariably greater in the hc*TIF1* vs. vector transformants.

4) In Figure 8C, the ded1-ts presumably already has defects compared to wild-type cells, especially in the stem-loop reporters. The authors should probably document these defects somewhere, even if they also show the relative differences as in Figure 8C. Along similar lines, in Figure 8—figure supplement 1, why are the ded1-ts values not all set to 1.0 like in Figure 8C? The legend indicates they should be the same as Figure 8C.

The defects in reporter expression conferred by *ded1-ts* are indicated in Figure 8—figure supplement 1, which has been cited explicitly in the Results; and the legend to Figure 8—figure supplement 1 has been modified to indicate that the results for the two SL reporters in the parental *ded1-ts* strain were not normalized to 1.0 in the manner conducted in Figure 8C.

5) Also regarding Figure 8C, the authors do not offer any interpretation of why the cap-proximal and cap-distal stem-loops are differentially affected by the different mutants. Why might this be?

As noted above for reviewer 1, we will add text to the Discussion regarding this point.

6) In Figure 8D, there does not appear to be any statistical analysis as to whether the observed changes are significant.

We have added a new supplementary figure (Figure 8—figure supplement 4) with evidence indicating statistically significant reductions in the proportions of the *FLUC* SL reporter found in heavy polysomes for both *ded1* alleles harboring substitutions in the NTD vs. the parental *ded1-ts* allele.

Reviewer #3 (Significance):This work advances our understanding of the mechanism of translation initiation, specifically the interactions of an RNA helicase, Ded1, that is known to be important in initiation, although its function has not been fully described. This study builds on previous work by this group and others (e.g. Hilliker et al., 2011; Senissar et al., 2014; Sen et al., 2015; Gao et al., 2016; Gupta et al., 2018) to better define the relationship of Ded1 and the eIF4F complex. As such, this work will be of significance to those in the translation field, particularly those studying initiation factors and/or RNA helicases. I expect that the identification of Ded1 binding determinants for eIF4A and eIF4E will be of marked interest for researchers studying these factors specifically. On a broad level, however, the research presented does not propose especially novel hypotheses or mechanisms and uses well-established techniques, so it is not likely to be appropriate for a high impact journal. This work is more of a "fill in the details" study, but it is solid work and should be published, pending the ability to address reviewer comments.

We respectfully disagree that this work merely “fills in details” as there was previously no compelling evidence that the individual interactions of eIF4A or eIF4E with Ded1 are important in vivo, nor that they are even more crucial when the Ded1CTD interaction with eIF4G is compromised. As such, the results provide the first strong evidence that robust Ded1 function in living cells is dependent on its association with the eIF4F complex, including separate contacts with each of its three subunits. Accomplishing this required a combination of genetics and biochemistry to map the locations of the distinct binding sites for eIF4A and eIF4E in the Ded1 NTD to the amino acid level, establish their importance for Ded1’s ability to enhance bulk and mRNA-specific translation in vivo and to stimulate 48S PIC assembly in a fully purified system; and also to demonstrate by suppressor analysis that the translation defects in vivo indeed arise from impaired association with eIF4A or eIF4E.